# A global exploratory comparison of country self-citations 1996-2019

**Alberto Baccini** [1]*, **Eugenio Petrovich** [2]

**1** Department of Economics and Statistics, University of Siena, Siena, Italy, **2** Department of Philosophy and Education Sciences, University of Turin, Turin, Italy

☯ These authors contributed equally to this work.

* alberto.baccini@unisi.it

## Abstract

Self-citations are a key topic in evaluative bibliometrics because they can artificially inflate citation-related performance indicators. Recently, self-citations defined at the largest scale, i.e., country self-citations, have started to attract the attention of researchers and policy-makers. According to a recent research, in fact, the anomalous trends in the country self-citation rates of some countries, such as Italy, have been induced by the distorting effect of citation metrics-centered science policies. In the present study, we investigate the trends of country self-citations in 50 countries over the world in the period 1996-2019 using Scopus data. Results show that for most countries country self-citations have decreased over time. 12 countries (Colombia, Egypt, Indonesia, Iran, Italy, Malaysia, Pakistan, Romania, Russian Federation, Saudi Arabia, Thailand, and Ukraine), however, exhibit different behavior, with anomalous trends of self-citations. We argue that these anomalies should be attributed to the aggressive science policies adopted by these countries in recent years, which are all characterized by direct or indirect incentives for citations. Our analysis confirms that when bibliometric indicators are integrated into systems of incentives, they are capable of affecting rapidly and visibly the citation behavior of entire countries.

## Introduction

Since the early times of citation indexes, self-citations have attracted the attention of bibliometricians [1, 2]. In evaluative bibliometrics, the main concern with self-citations is that they can potentially inflate impact metrics or distort their meaning [3–6]. It has thus been debated whether they should be removed from citation indicators [7–10]. In descriptive bibliometrics, on the other hand, self-citations have been studied from the point of view of scholarly communication. The motivations for self-citing have been classified [11, 12] and self-citations have been used to investigate how scientific authors relate with their own production [13, 14].

In the most general sense, self-citation occurs when an entity (e.g., an author, journal, institution, or country) receives a citation from a publication produced by the same entity [15]. Even if all self-citations derive from the act of self-referencing, self-citations and self-references should be distinguished [16]. For technical reasons shortly described below, the results derived

**Data Availability Statement:** All relevant data are within the manuscript and its Supporting information files.

**Funding:** Alberto Baccini acknowledges the funding by the Italian Ministry of University, PRIN project: 2017MPXW98.

**Competing interests:** NO authors have competing interests.

from observing self-citations or self-references are different and so is their interpretation. This paper focuses on country self-citations.

Depending on the entity considered, self-citation can be classified into different types [6]. The most basic is the *author self-citation*, which occurs when the publications written by an author are cited in the following publications by the same author. For multi-authored publications, author self-citation can be defined narrowly or broadly, i.e., including or not citations generated by co-authors. Author self-citations intended in the extensive sense are sometimes called co-author self-citations [17] or all-author to all-author self-citations [6]. *Journal self-citation* occurs when an article published in a certain journal is cited by a subsequent publication in the same journal [18]; this form of self-citation has been mainly studied to understand how it can influence or even manipulate the journal Impact Factor [19–21]. *Institution self-citation* happens when the authors of the cited and of the citing publications share the same affiliation [22, 23]. By extending rather inappropriately the notion of self-citation, *language self-citation* refers to citations occurring among publications in the same language [24], and *field self-citation* for citations between publications belonging to the same academic field [25].

In the context of the study of the scientific performance of countries, the self-citations defined at the highest level of aggregation, i.e., *country self-citations*, have recently started to attract some attention, both from researchers [26–31] and in science policy reports [32]. A country self-citation, also called sometimes domestic citation (e.g., [33]), occurs when the publications produced by the researchers of a country are subsequently cited by researchers of the same country.

In this study, the trends of country self-citations in 50 countries worldwide are investigated in order to reveal groups of countries characterized by similar self-citation behavior in time. The main interest consists in individuating countries that *deviate* from standard trends because these anomalies may signal perturbations in the scientific development possibly induced by science policies. Italy is a case in point in this sense, as previous research has revealed an anomalous rise in the country's self-citations after the introduction of pervasive bibliometric evaluation [26, 34–38].

The paper is structured as follows. In the next section, the bibliometric indicators based on country self-citations are reviewed. Then, the indicators, data, and analytical methods used in this study (time-series analysis) are presented. The main findings are shown in the Results section, whereas, in the Discussion section, we focus on countries characterized by anomalous self-citation trends. Based on the detailed reconstruction of the research policies adopted by these countries, it is argued that the anomalous trends are most likely explained by the adaptive response of scientists to the systems of incentives established by the policies themselves. Accordingly, Conclusions suggest managing bibliometric indicators in research policy contexts with extreme caution.

## Review of the main indicators based on country self-citations

As anticipated, self-citations and self-references are defined, computed, and interpreted in different ways. To highlight these differences [39], for example, called the former diachronous or prospective self-citations and the latter synchronous or retrospective self-citations.

Self-citations can be defined as citations from citing sources to cited items that are both produced by (at least) the same entity *E*, i.e. the same author, journal, institution, or country. Different ways of defining these entities entail the use of different algorithms, which will be discussed in the next paragraph.

The computation of self-citations requires the definition of: (i) a publication window delimiting the cited items as the ones published in it. For the sake of simplicity, in what follows, the

publication window is set at one year, hence the cited items considered are only the ones published in the year $y$; and (ii) a citation window delimiting the citing sources as the ones published in it. Let $y$ be the reference year for calculation and also the publication window; $T$ is the length of the observation period, expressed in years. Self-citations of an entity $E$ in the year $y$ can be defined as:

$$S_{E,y} = \sum_{i=k}^{T} s_E(y, y - 1 + i),$$ (1)

where $s_E(y, y - 1 + i)$ is the number of self-citations received by the set of cited items published by the entity $E$ in the publication window $y$, from citing sources produced by the same entity $E$ in the citation window $(y - 1 + i)$. The citation window includes the years $(y - 1 + i)$ for $i = k, \ldots, T$, where $k \in \{1, \ldots, T\}$ is chosen by the user for setting the desired time interval for observing citations. If $k > 1$ citation window and publication window are disjoint; for $k = 1$ and $T > 1$ they are partially overlapped, and for $k = T = 1$ they are completely overlapped. In the first case, citing sources are published in years following the publication windows; in the second case, the citation window includes the publication window and the following years; in the third one, citation window and publication window coincide. Analogously, the total citations received by the set of cited items published by the entity $E$ in the year $y$ can be defined as:

$$C_{E,y} = \sum_{i=k}^{T} c_E(y, y - 1 + i)$$ (2)

where $c_E(y, y - 1 + i)$ is the number of citations received by the set of cited items published by $E$ in the publication window $y$, from citing sources published in the citation window $y - 1 + i$.

The basic indicator is the *self-citation rate* ($SR_E$, $y$) for entity $E$ in the year $y$, defined as the ratio between the self-citations and the total number of citations received by $E$ [26, 28, 40, 41]:

$$SR_{E,y} = \frac{S_{E,y}}{C_{E,y}} = \frac{\sum_{i=k}^{T} s_E(y, y - 1 + i)}{\sum_{i=k}^{T} c_E(y, y - 1 + i)},$$ (3)

where $SR_{E,y} \in [0, 1]$. $C_{E,y}$ is usually interpreted as a proxy of the academic impact of the entity $E$. $S_{E,y}$ is an indicator of academic impact generated by self-citations of $E$. Hence, a self-citation rate can be interpreted as the proportion of the academic impact of $E$ due to its self-citation activity. It should be noted, that the denominator of the ratio is concretely produced by the citing choices of the whole scholarly community, and the ratio relates the self-citing choices of $E$ with respect to the citing choices of the whole scholarly community.

Self-references indicate as well citations from citing sources to cited items that are both produced by (at least) the same entity, but the sets of citing sources and cited items are defined in different ways with respect to self-citations. Moreover, the computation of self-reference requires only a publication window for the set of citing sources. If the publication window is set also at one year, the cited sources are the publications produced by $E$ in the year $y$, and cited items are (all or part of) the items listed in the bibliographies of the citing sources. $SB_{E,y}$, i.e. the number of self-references in the year $y$, is computed by summing up the citations from citing sources published in the year $y$ by $E$ to cited items previously produced by $E$. Analogously, $R_{E,y}$ indicates the total number of citations given (i.e., references) by $E$ in the year $y$ to

documents published before. The self-reference rate $SBR_{E,y}$ for the year $y$ is thus defined as:

$$SBR_{E,y} = \frac{SB_{E,y}}{R_{E,y}}. \tag{4}$$

Note that a rigorous definition of this indicator would require the specification of a reference window, analogous to the citation window used in the self-citation rate. The studies based on the self-reference rate (e.g., [16]) usually assume a reference window that is extended to the oldest publication year of the references considered, but, in principle, other reference windows are possible.

When references are observed, the focus of the analysis is on the use of previous knowledge by the entity $E$, and not on the academic impact of the scientific production of $E$. It should be noted that the denominator of the ratio is, in this case, produced also by the citing choices of $E$. $SBR_E$ should be interpreted as an indicator of how much $E$ uses in its current work the knowledge that it had previously produced. For instance, for an author, a relatively high value of $SBR_{E,y}$ may indicate that her/his work is largely based on her/his previous work. This in turn may be due to scarce attention to the work of other researchers in the field or to the fact that the author works in a little and specialized niche [39]. A relatively high value of $SR_{E,y}$, instead, indicates that a relatively high share of her/his academic impact is due to her/his self-citation activity.

Note that, in general, authors have more control over their self-reference rate than their self-citation rate, as they can manage more easily the citations they give (i.e., the denominator of the self-reference rate), than the citations that they receive (i.e., the denominator of the self-citation rate). In fact, authors interested in artificially inflating their citation metrics can effectively conceal strategic self-references within long reference lists and keep their self-reference rate low. Still, if they are not able to attract enough external citations to offset the impact of their strategic self-references, their self-citation rate will inevitably rise. This shows that the self-citation rate is more effective in revealing strategic self-citation behavior than the self-reference rate.

The basic indicator in the literature on country self-citations is the *country self-citation rate*. Hereafter $N$ denotes the entity "country"; the self-citation rate of country $N$ in the year $y$ is the ratio between its country self-citations and the total number of citations received by that country [26, 28, 40, 41]:

$$SR_{N,y} = \frac{S_{N,y}}{C_{N,y}} \tag{5}$$

where $S_{N,y}$ is the raw number of self-citations of country $N$ in the year $y$ and $C_{N,y}$ is the total number of citations received by $N$ in the year $y$.

Computing the total of country self-citations for a country $S_{N,y}$ is, however, less obvious than it may seem. As detailed in Methods and data, country self-citations can be in fact computed either narrowly or broadly, depending on how citations to international publications are considered. In [26], a variant of the self-citation rate indicator called "inwardness" is developed, characterized by a citation window variable in length. A *broad* definition of country self-citation is adopted, according to which a citation is considered a country self-citation when the intersection between the set of the countries of affiliation of the author(s) of the cited publication, and the set of the countries of affiliation of the author(s) of the citing publication, is not empty. This broad definition of country self-citation has the desirable property of ascribing to the world an inwardness value of inwardness 1, which makes therefore the inwardness an indicator normalized for the size of the country in terms of publications (see [26] for details). This

property, however, does not hold for the $SR_{N,y}$ when the country self-citations are computed narrowly (see Sec. Methods and data).

Inwardness is interpreted as a measure of the self-referentiality of a country: a higher level of inwardness suggests that the scientific publications produced by a country attract mainly the interest of the national community, whereas a lower level of inwardness suggests that the scientific production is cited mainly abroad. In the same sense, [42] related country self-citations to the degree of scientific insularity of nations. [42] also suggested several factors that may explain why developing countries show higher self-citation rates, among which a focus on applied scientific issues that respond to the perceived needs of national development, poor referencing practice, insufficient training of graduate students, preference for literature in national language than in English, and the proliferation of low-quality national journals. As [26] have shown, however, also developed countries may show anomalous raises in self-citation rates induced by research evaluation policies that reward raw citation metrics.

A variant of the self-citation rate is the citation domesticity indicator [43], in which citations coming from international publications are evenly distributed among the citing countries. Apart from [43], however, fractional counting of country self-citations has never been considered in the following literature. This may be explained by the fact that most more recent studies collect data on country self-citations from Elsevier's SCImago or SciVal platforms, which do not apply fractional counting [44].

The complement of the self-citation rate is the *foreign citation rate $FR_{N,y}$*, also called "international scholarly impact of scientific research" by [45]:

$$FR_{N,y} = \frac{C_{N,y} - S_{N,y}}{C_{N,y}} = 1 - SR_{N,y} \tag{6}$$

The country self-citation rate results to be positively correlated with the publication output of a country. In particular, [30] proposed a model where the country self-citation rate increases with the logarithm of the output:

$$SR_{N,y} \propto \log P_{N,y} \tag{7}$$

where $P_{N,y}$ is the number of publications of country $N$ in the year $y$. This occurs because bigger countries have more domestic papers to cite and, hence, are more likely to attract citations from their own researchers than smaller countries [40, 41, 46]. By contrast, the average number of citations per document of a country is negatively correlated with self-citation rates [29]. Self-citation rates have increased over time: according to [29] estimates, the average self-citation rate of 62 countries raised of 28.9% from 1996 to 2009. [26] noted as well that the Inwardness of G10 countries increased during the period 2000–2016 with a mean increase of 5.2 percentage points.

To correct from the size-dependency of self-citation rates, [40] proposed to compare the self-citation rates with the world-share of publication of a country, based on the idea that if the publications from a country are cited as expected, then its share of country self-citations is proportional to its share of world publications. More recently, this indicator has been called "over-citation ratio" ($OCR_{N,y}$) by [27] and is defined as:

$$OCR_{N,y} = \frac{\dfrac{S_{N,y}}{C_{N,y}}}{\dfrac{P_{N,y}}{P_{w,y}}} = \frac{SR_{N,y}}{\alpha_{N,y}} \tag{8}$$

where $P_{w,y}$ is the total number of publications in the world and $\alpha_{N,y}$ the proportion of

publications of country $N$ in the world. An over-citation ratio higher than 1 means that the country receives more citations from its own publications than expected based on its relative weight in the world scientific production. At the field level, [27] found that, as expected, over-citation ratio is higher for scientific fields of more national interest.

However, the over-citation ratio results to be size-dependent, as it is highly influenced by the denominator in the formula, i.e., the fraction of papers published by a country. For countries with high weight in the world publication output, such as the USA, the $OCR_{N,y}$ will be always smaller than for small countries. For instance, for a country publishing one-third of the papers in the world ($\alpha_{N,y}$ = 33%), the $OCR_{N,y}$ can never exceed a value of $1/0.33 \approx 3$, whereas a small country that published $\alpha_{N,y}$ = 3.3% of world papers, the maximum value of the $OCR_{N,y}$ raises to 30 [24, 41]. In fact, [27] found that there is a negative power correlation between $OCR_{N,y}$ and $\alpha_{N,y}$:

$$OCR_{N,y} \propto \frac{1}{\alpha_{N,y}^J}. \tag{9}$$

A further indicator, based on probability ratio, has thus been proposed, first in the study of language self-citations by [46] and then adapted to country self-citations by [41]. This indicator, called "odds-ratio" ($O_{N,y}$), relates two ratios: the numerator is the ratio of country self-citations to foreign citations, and the denominator is the ratio of domestic publication proportion to foreign publication proportion:

$$O_{N,y} = \frac{SR_{N,y}/(1 - SR_{N,y})}{\alpha_{N,y}/(1 - \alpha_{N,y})}. \tag{10}$$

The odds-ratio $O_{N,y}$ measures to what extent the country relative preference to cite its own publications is greater or smaller than the existing ratio of its domestic publications to publications from other countries. Note that for small values of $SR_{N,y}$ and $\alpha_{N,y}$, the odds-ratio approaches the over-citation ratio.

The odds-ratio has three drawbacks as well, however [24]. First, if a country cites only its own publication, the measure is infinite. Second, it is oversensitive to small variations in the $SR_{N,y}$. Third, it is not normalized between 0 and 1. To fix these issues, [24] proposed the following indicator of relative self-citation rate:

$$E_{N,y} = SR_{N,y} \ln\left(\frac{1}{\alpha_{N,y}}\right). \tag{11}$$

According to their interpretation, this formula considers the publication proportion $\alpha_{N,y}$ as a stimulus to the publication-citation system and the relative self-citation rate $E_{N,y}$ as the subjective reaction of the system, which depends logarithmically on the intensity of the stimulus, as in Weber-Fechner equation. The function also expresses the law of diminishing returns: the larger $\alpha_{N,y}$, the less important the changes in the relative self-citation rate. The relative self-citation rate has the advantages of being normalized and size-independent. Besides its mathematical merits, however, its meaning is less transparent compared to all the previous alternatives.

## Methods and data

As anticipated, this work handles time series analysis of country self-citations: time-series clustering techniques are used for detecting countries whose self-citation behavior is similar [47, 48]. To build the time-series, a preliminary distinction between extensive (or broad) and restrictive (or narrow) country self-citations is introduced. The two counts are conceptually

different and generate different estimates of country self-citations. Hence, two self-citation indicators are defined.

The relevant data from a citation database for the countries of interest are next retrieved and the time-series generated. The distance between the time-series is then calculated using a suitable distance measure and the structure of the distance matrix thus obtained is explored using multi-dimensional scaling (MDS). In the resulting MDS maps, countries characterized by similar trends will be placed close to each other, whereas countries with different trends far away [49].

Fig 1 sums up the phases of the present study. In the next paragraphs, the methodological and technical choices taken in each step are presented in detail and justified.

## Country self-citations of Type I and II

As said above, country self-citations can be computed in different ways depending on how self-citations to international publications, i.e. publications with authors from different countries, are considered. Analogously to author self-citations, the count of country self-citations for international publications can be done by adopting a *publication-based* or a *author-based* perspective [34].

The extensive publication-based perspective, adopted in the Inwardness indicator and implemented in the SciVal database, considers as country self-citations *all* citations coming from the collaborating countries. In the following, these extensively-intended country self-citations are referred to as country self-citation of Type I ($SR^I$), by omitting for simplicity the indexes of the publication window $y$ and of the country $N$. The second way of counting country self-citations is author-based and more restrictive. Country self-citations of Type II ($SR^{II}$) are computed by considering only national *author* self-citations: a publication produced by (at least) an author of a given country receives a country self-citation of Type II if the citing publication is authored by one of the authors of the cited one, and this author is affiliated with the considered country.

The example in Fig 2 clarifies the computation of the two types. The Figure shows a citation network comprising publications labeled as 1, 2, . . ., 8, of which 3 are cited items and 5 are citing sources, and 8 citations indicated by the arrows linking the pairs of cited and the citing publications: (1, 4), (1, 5), . . ., (3, 8). The publications are authored by 5 authors labeled *A*, *B*, *C*, *D*, *E* from 3 countries: Italy (IT), Netherlands (NL) and Canada (CA). Inside each node, the authors of the relative publication and their countries of affiliation are reported by letters and acronyms. Three statistics for the two cited countries (Italy and Netherlands) are then calculated (Table 1): the total number of country citations, the total number of country self-citations Type I, and the total number of country self-citations Type II. For country citations, it is sufficient to count the citations landing on the publications produced by each country: the 3 Italian publications get 8 citations and the 1 Dutch publication 5 citations. For country self-citations Type I, we must compare for each citation the set of countries of the citing publications with that of the cited publications: when the intersection is not empty, the citation counts as a country self-citation Type I. Thus, Italy collects 7 country self-citations Type I, the Netherlands 4. Note, however, that the Dutch-Italian publication 1 receives only 1 citation from a publication with a Dutch author (publication 5), whereas the other three citing publications 6, 7, and 8 are in fact from Italian authors. This happens because country self-citations Type I of international publications include the citations coming from *any* of the collaborating country. In fact, a paper resulting from the collaboration of numerous countries is more likely to attract self-citations of Type I in comparison to a paper resulting from the collaboration of only a few countries. This is because there is a higher likelihood that the countries of the citing papers will

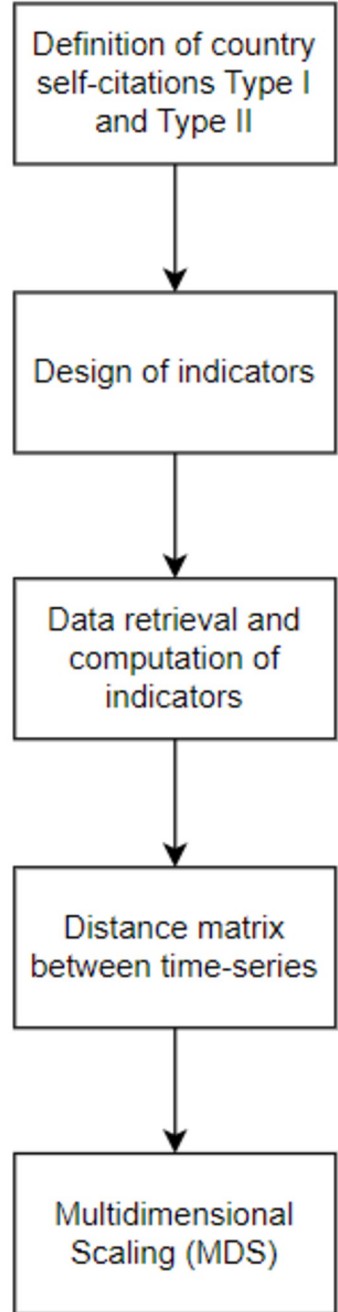

**Fig 1. Phases of the analysis.**

overlap with the countries of the former rather than with the countries of the latter (see [26], sec. 3). In the extreme hypothetical scenario of a paper resulting from the collaboration of all countries in the world, any citation to this paper would be considered a self-citation of Type I, as the country of the citing papers will always overlap with at least one of the countries of the cited paper.

Country self-citations of Type II are introduced precisely to correct this somehow counter-intuitive property of country self-citations Type I. Country self-citations of Type II in fact

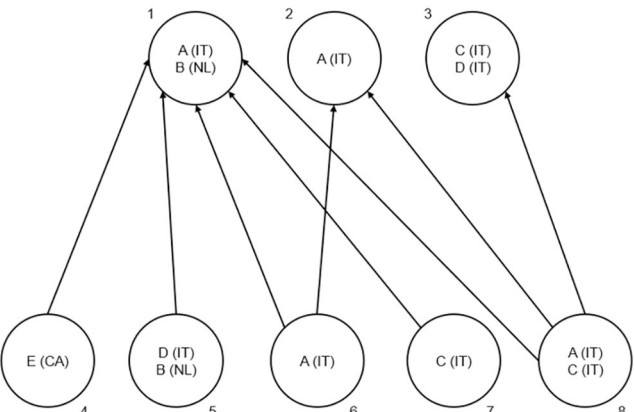

**Fig 2. Toy citation network.** Nodes represent publications and arrows citation links. The authors of each publication, represented by capital letters, and their affiliation country, represented by acronyms, are shown inside each publication node.

include only the citations coming from the focal country. Then, if the focal country is the Netherlands, only the citation-link (1, 5) in the example will count as a country self-citation Type II for the Netherlands because it links two publications sharing the same Dutch author *B*. Citations (1, 6) and (1, 8), by contrast, will not count because they are generated by the Italian co-author of *B*. Symmetrically, citation (1, 5) will not count for Italy, whereas (1, 6) and (1, 8) will. Table 1 shows analytically in which relationship each citation-link stands with the two countries and reports the three statistics for each country. Note that no fractional counting was applied: citations from multi-author or international publications or landing on multi-author or international publications are not divided among the cited or citing countries.

Based on the previous definitions, the difference between the country self-citations of Type I and Type II of a country should be attributed mainly to the publications the country produces with other countries, i.e., to its international publications. A wider difference means that these international collaborations are widely cited not only by the focal country but also by the collaborating countries, increasing Type I but not Type II self-citations. A shorter difference, by contrast, means that the country has few international collaborations and/or that they do

**Table 1. Classification of citations in Fig 2 based on the two definitions of country self-citation.** No fractional counting is applied.

| | Italy: | | | Netherlands: | | |
|---|---|---|---|---|---|---|
| | 3 cited publications | | | 1 cited publication | | |
| | 4 citing publications | | | 1 citing publication | | |
| Link | Citation | Self-cit Type I | Self-cit Type II | Citation | Self-cit Type I | Self-cit Type II |
| 1–4 | X | | | X | | |
| 1–5 | X | X | | X | X | X |
| 1–6 | X | X | X | X | X | |
| 1–7 | X | X | | X | X | |
| 1–8 | X | X | X | X | X | |
| 2–6 | X | X | X | | | |
| 2–8 | X | X | X | | | |
| 3–8 | X | X | X | | | |
| Tot | 8 | 7 | 5 | 5 | 4 | 1 |

not attract citations from the collaborating countries. The difference between Type I and Type II is therefore related to the international citation impact of a country.

Part of the difference between the country self-citations of Type I and Type II of a country can be attributed also to citations generated by country publications that are not also author self-citations. In particular, the country self-citations of Type II do not capture citation exchanges between groups of national authors who cite each other but do not collaborate directly on writing papers.

## Indicator design

The review of the literature above presented several indicators based on country self-citations. However, only the self-citation rate *SR* is a pure citation indicator. The others integrate in different ways publication counts to correct for the size-dependency of the *SR*.

In static analyses of country self-citations, indicators encapsulating both publications and country self-citations are useful, but they become problematic when the temporal dimension is considered. In fact, the trend of the publications-plus-self-citations indicators results to be affected both by the publication trend and the self-citation trend. The indicators, however, cannot say which of them is the driver. Consider, for instance, the *over-citation ratio*: an increasing over-citation ratio over time may be caused both by a raise in the self-citation rate with the share of country publications remaining stable or by a decrease in the share of country publications with the self-citation rate remaining stable. The indicator as such does not say which of the two dynamics has taken place. Moreover, the share of country publications depends on the publication activities of other countries as well: a country experiencing a rise in scientific productivity, like China, causes an automatic reduction in the publication shares of the other countries, even if their productivity has remained steady. Therefore, too many dynamics affect the trends of the over-citation ratio. An analogous reasoning applies to the other mixed indicators.

Since the focus of this paper is on country self-citation dynamics, the most suitable indicator is the *self-citation ratio* only. Specifically, two variants of the indicator will be computed: one based on country self-citations Type I (*extensive* self-citation ratio, $SR^I$) and one based on country self-citations Type II (*restrictive* self-citation ratio, $SR^{II}$). Even if country self-citations of Type I are affected by citations coming from other countries, as seen above, they remain interesting because they allow capturing, among other things, the effect of *citation clubs* at the country level, i.e., the strategic exchanges of citations between researchers in the same country that are not co-authors [26]. The $SR^I$, like the Inwardness, is a size-independent measure of country self-citation, since it is normalized for the size of country publications; $SR^{II}$ is instead not normalized.

The last step in the design of the indicator is the definition of an appropriate *citation window*. A citation window is required to correct for the fact that older publications have more time to accumulate citations. In evaluative bibliometrics, longer citation windows are sometimes recommended to capture the delayed impact of publications [50]. For our study, however, a long citation window has two shortcomings. The first one is that the larger the citation window, the more observations are lost in the time series since, for the more recent years, the citation window is not complete.

The second one is a smoothing effect on perturbations. Imagine an anomalous peak in country self-citations due to a nationwide policy change, which determines an amount of 80 country self-citations Type II at year $y_0$. Imagine that the policy is afterward dismissed so that the country self-citations return to "normal" values of 40 in the following years $y_1$ and $y_2$. Lastly, stipulate that the country total citations have remained stable at 100 for all three years.

Now, if we use a 1-year citation window, i.e., we count only the citations coming from publications that appeared in the same year of the cited publications, the $SR^{II}$ for the three years will result to be, respectively, 0.8, 0.4, and 0.4. With this citation window, the peak at year $y_o$ is clearly visible. With a 2-year citation window, which sums together the self-citations of the cited years and those of the next year, by contrast, the values will be 0.6 and 0.4 (note the value for cited year $y_2$ can no longer be calculated). Finally, with a 3-year citation window, the $SR^{II}$ at year $y_0$ will result to be 0.53 (no further values can be computed). The aggregation of the anomalous year with the normal ones has reduced the visibility of the anomaly. With a sufficiently long citation window, the anomaly might even disappear, being absorbed by the average trend.

The yearly Type I or Type II self-citation ratios for country $N$ in the year $y$ are computed by setting a 2-year citation window as follows:

$$SR_N = \frac{S_{N,y}}{C_{N,y}} = \frac{\sum_{i=1}^{2} s_N(y, y-1+i)}{\sum_{i=1}^{2} c_N(y, y-1+i)}. \tag{12}$$

Therefore, for a country $N$, the number of citations and self-citations Type I/Type II collected in a given year $y$ are computed by considering citations and self-citations Type I/Type II received in the year $y$ by the cited items published in the year $y$ from the citing sources published in the years $y$ and $y + 1$. This choice permits reducing at a minimum of one year the loss of observations in each time-series; moreover, it permits limiting the smoothing effect and highlighting anomalies in trends. To check for the effect of the citation window, i.e., to check whether the final results were affected by the choice of the citation window, all the analyses were repeated by using also 1-year and 5-year citation windows, by mimicking the usual citation windows of journal impact factors [51]. These results, which are qualitatively similar to those reported in the paper, can be found in the Supporting Information.

## Data

Scopus data were used for the computation of the indicators and data were provided by Elsevier through its ICSR Lab. The identification of author self-citations used in the calculation of Type II country self-citations relies on the Scopus author profiling procedure described in [52].

Countries with at least 100,000 publications indexed in the Scopus databases in the period 1996–2019 were considered ($n = 50$). The indicators were computed for each country on a yearly basis from 1996 to 2019 using, as said above, a 2-year citation window. Publications from all Scopus fields were aggregated, i.e., for each country, the entire scientific output was considered, with no distinction of the research area. Thus, for each indicator, 50 time-series (1 per country) with 24 observations each (1 per year) were computed. In the Supporting Information, the trends of the *over-citation ratio*, *odds-ratio*, and *relative self-citation rate* are available as well. All were calculated using both Types of country self-citations.

Tables 2 and 3 provides the descriptive statistics of the countries in the dataset.

## Comparing time-series

In the literature on time-series analysis, numerous measures for comparing time-series have been developed (reviews can be found in [47, 48, 53]). The various measures encapsulate different senses in which two time-series may be similar or dissimilar. Choosing the suitable measure depends both on the nature of the data and the purposes of the analysis.

**Table 2. Descriptive statistics 1996–2019.** The overline indicates the mean. Publications include only research articles, reviews, and conference papers. Citations ($\overline{C}$) and self-citations ($\overline{S^I}$ and $\overline{S^{II}}$) are computed on a 2-year citation window. Citing sources include all types of documents, cited publications include research articles, reviews, and conference papers.

| Country | Publications | International Publications % | $\overline{C}$ | $\overline{S^I}$ | $\overline{S^{II}}$ |
|---|---|---|---|---|---|
| United States | 11,414,720 | 27.5 | 1,464,481.7 | 797,739.3 | 306,809.7 |
| China | 6,479,473 | 17.9 | 504,359.4 | 317,813.6 | 114,974.0 |
| United Kingdom | 3,153,786 | 43.6 | 419,930.0 | 130,947.8 | 79,121.6 |
| Germany | 2,947,845 | 42.5 | 376,685.1 | 129,782.3 | 86,071.0 |
| Japan | 2,788,160 | 21.4 | 231,987.1 | 85,635.3 | 63,336.1 |
| France | 2,064,932 | 45.6 | 245,978.8 | 71,137.5 | 50,673.5 |
| India | 1,733,461 | 16.6 | 112,886.3 | 45,132.6 | 29,494.7 |
| Italy | 1,711,410 | 39.2 | 213,541.0 | 69,307.8 | 50,916.5 |
| Canada | 1,690,621 | 43.7 | 216,283.8 | 54,260.4 | 40,718.7 |
| Spain | 1,364,243 | 40.4 | 157,716.2 | 45,689.5 | 33,642.1 |
| Australia | 1,320,648 | 44.3 | 171,197.2 | 47,861.2 | 35,692.6 |
| Russian Federation | 1,167,226 | 25.9 | 60,725.2 | 26,180.6 | 20,718.6 |
| South Korea | 1,158,182 | 25.9 | 105,930.1 | 30,022.7 | 22,081.7 |
| Brazil | 976,69 | 27.8 | 69,322.7 | 23,589.9 | 16,348.8 |
| Netherlands | 935,681 | 50.7 | 149,365.2 | 34,158.1 | 26,825.5 |
| Switzerland | 694,479 | 60.0 | 123,581.9 | 25,734.6 | 20,675.7 |
| Poland | 682,376 | 29.3 | 53,484.4 | 17,605.0 | 13,987.6 |
| Taiwan | 672,686 | 22.4 | 54,105.9 | 14,520.2 | 11,800.8 |
| Sweden | 646,746 | 52.3 | 95,146.9 | 20,784.7 | 17,090.6 |
| Turkey | 597,155 | 19.1 | 38,182.5 | 10,944.1 | 7,591.9 |
| Iran | 563,672 | 20.8 | 46,282.9 | 19,845.8 | 14,466.0 |
| Belgium | 521,363 | 55.7 | 77,294.0 | 16,065.7 | 14,232.2 |
| Denmark | 386,678 | 54.5 | 64,885.1 | 13,284.3 | 11,391.4 |
| Austria | 380,443 | 54.7 | 52,936.2 | 10,928.6 | 9,766.2 |
| Israel | 368,383 | 42.2 | 47,940.3 | 9,543.1 | 8,384.1 |
| Czech Republic | 339,348 | 37.9 | 30,120.1 | 8,700.5 | 7,332.9 |
| Finland | 335,323 | 47.9 | 45,315.2 | 10,208.1 | 8,925.1 |
| Mexico | 326,559 | 39.4 | 25,077.1 | 6,014.5 | 4,895.2 |
| Hong Kong | 311,764 | 59.3 | 38,812.4 | 8,124.7 | 7,515.1 |
| Malaysia | 310,874 | 35.5 | 21,162.5 | 7,031.0 | 5,714.8 |
| Greece | 306,955 | 41.5 | 33,923.5 | 7,385.5 | 6,482.3 |
| Portugal | 306,884 | 48.3 | 34,792.4 | 8,413.6 | 7,636.2 |
| Norway | 305,437 | 51.7 | 40,642.6 | 8,923.3 | 7,451.2 |
| Singapore | 292,723 | 51.5 | 41,668.3 | 8,539.5 | 7,830.5 |
| South Africa | 276,641 | 42.9 | 27,013.8 | 7,259.7 | 5,606.1 |
| New Zealand | 231,225 | 48.9 | 26,678.4 | 5,947.3 | 5,053.0 |
| Egypt | 221,805 | 42.1 | 17,146.1 | 4,617.4 | 3,928.0 |
| Argentina | 211,228 | 40.7 | 19,398.9 | 4,275.7 | 3,619.9 |
| Romania | 210,962 | 32.2 | 14,083.3 | 4,307.6 | 3,490.0 |
| Ukraine | 201,738 | 34.1 | 10,129.3 | 3,686.2 | 3,290.4 |
| Saudi Arabia | 201,523 | 63.8 | 25,274.4 | 5,862.0 | 5,302.3 |
| Ireland | 196,413 | 51.6 | 27,564.5 | 4,974.7 | 4,639.1 |
| Hungary | 193,999 | 44.9 | 20,569.6 | 4,395.8 | 3,796.7 |
| Thailand | 188,262 | 39.6 | 14,465.1 | 3,219.8 | 2,706.9 |
| Pakistan | 168,222 | 42.4 | 14,633.1 | 4,746.5 | 3,640.3 |

*(Continued)*

**Table 2.** (Continued)

| Country | Publications | International Publications % | $\overline{C}$ | $\overline{S^{I}}$ | $\overline{S^{II}}$ |
|---|---|---|---|---|---|
| Chile | 154,667 | 54.9 | 17,123.8 | 3,908.5 | 3,130.5 |
| Indonesia | 150,879 | 27.9 | 6,574.9 | 2,575.6 | 1,908.5 |
| Slovakia | 115,855 | 41.3 | 8,905.9 | 2,378.4 | 1,866.2 |
| Croatia | 107,419 | 31.5 | 7,925.5 | 1,765.6 | 1,523.6 |
| Colombia | 107,333 | 47.5 | 9,269.6 | 1,814.2 | 1,470.0 |

In the present setting, the measure should satisfy three conditions. First, it should *not* be sensible to the mere magnitude of the difference between the self-citation ratios, since self-citation ratios are partly affected by the size of the country: bigger countries tend to have higher self-citation ratios. This excludes all measures based on the point-wise distance between the time-series. Second, the measure should be sensitive to *trends* and *changes in trends* of self-citation ratios, as these events may be associated with external perturbations, such as policy changes, useful in explaining the phenomenon. Third, the measure should not assume any underlying statistical model for self-citation ratios, in order to avoid unjustified assumptions on the dynamics of the self-citations over time.

The following dissimilarity measure based on Pearson's correlation satisfies all the three requirements specified above:

$$d(\boldsymbol{X}, \boldsymbol{Y}) = \sqrt{2(1 - \rho(\boldsymbol{X}, \boldsymbol{Y}))} \tag{13}$$

where $\boldsymbol{X} = (X_1, \ldots, X_n)$, $\boldsymbol{Y} = (Y_1, \ldots, Y_n)$ are the time-series considered and $\rho(\boldsymbol{X}, \boldsymbol{Y})$ is the Pearson correlation index.

This measure was proposed originally by [54] and implemented in the function `diss.COR` of the package **TSclusts** [53] for R [55]. The measure is bounded between 0, when there is a perfect correlation between the time-series, and 2, where there is perfect anti-correlation between them. When the two series show no correlation at all, the value is $\sqrt{2}$.

All the dissimilarities between pairs of country trends can be arranged in a dissimilarity matrix. This in turn can be visualized using Kruskal's Non-metric Multidimensional Scaling, one form of non-metric MDS which respects the ranking of dissimilarities rather than their absolute values [49]. The function `isoMDS` of the package **MASS** [56] in R can be used to produce MDS maps.

## Results

Fig 3 shows the trends of the 50 countries for the two indicators $SR^I$ and $SR^{II}$. Three main observations can be made on these trends. First, the proportion of country self-citations of both types has decreased over time in most countries, following a linear pattern with more or less pronounced oscillations depending on the country. Three countries, however, deviate from this general behavior: in the case of Indonesia, Ukraine, and the Russian Federation, in fact, the trends of both indicators show an inversion from descending to ascending.

Second, the $SR^I$ and $SR^{II}$ trends are highly correlated for most of the countries, with both indicators following similar trajectories over time. Again, there is a notable exception represented by China, which is the only country where the share of country self-citations of Type I has surged (China $SR^I$ has increased by 24.1 p.p. between 1996 and 2018) while that of self-citations of Type II has substantially contracted ($SR^{II}$ has decreased by 22 p.p. in the same period).

**Table 3. Average country self-citations of type I and type II and their average difference.** All values are multiplied by 100.

| Country | $\overline{SR^{I}}$ | $\overline{SR^{II}}$ | $\overline{SR^{I}} - \overline{SR^{II}}$ |
|---|---|---|---|
| Argentina | 25.6 | 22.7 | 3.0 |
| Australia | 30.2 | 23.6 | 6.6 |
| Austria | 22.9 | 21.0 | 2.0 |
| Belgium | 23.2 | 21.1 | 2.1 |
| Brazil | 36.6 | 27.8 | 8.8 |
| Canada | 26.6 | 20.6 | 6.1 |
| Chile | 24.8 | 21.0 | 3.8 |
| China | 57.0 | 30.5 | 26.5 |
| Colombia | 21.0 | 18.9 | 2.0 |
| Croatia | 28.9 | 25.4 | 3.6 |
| Czech Republic | 32.6 | 28.4 | 4.1 |
| Denmark | 22.6 | 19.8 | 2.8 |
| Egypt | 30.1 | 27.4 | 2.7 |
| Finland | 25.0 | 22.1 | 2.9 |
| France | 30.5 | 22.1 | 8.5 |
| Germany | 36.0 | 24.4 | 11.6 |
| Greece | 25.1 | 22.8 | 2.3 |
| Hong Kong | 26.7 | 25.3 | 1.4 |
| Hungary | 24.3 | 21.7 | 2.6 |
| India | 41.9 | 31.1 | 10.8 |
| Indonesia | 26.1 | 21.6 | 4.5 |
| Iran | 45.8 | 38.6 | 7.2 |
| Ireland | 19.9 | 19.1 | 0.8 |
| Israel | 21.7 | 19.4 | 2.3 |
| Italy | 33.1 | 25.0 | 8.1 |
| Japan | 38.5 | 28.7 | 9.8 |
| Malaysia | 34.1 | 30.2 | 4.0 |
| Mexico | 27.1 | 23.1 | 4.0 |
| Netherlands | 24.8 | 20.0 | 4.8 |
| New Zealand | 25.4 | 21.7 | 3.7 |
| Norway | 24.9 | 21.1 | 3.8 |
| Pakistan | 35.9 | 30.1 | 5.7 |
| Poland | 35.7 | 29.7 | 6.1 |
| Portugal | 28.2 | 26.6 | 1.6 |
| Romania | 34.4 | 31.1 | 3.2 |
| Russian Federation | 41.1 | 34.3 | 6.9 |
| Saudi Arabia | 26.1 | 24.6 | 1.5 |
| Singapore | 26.3 | 25.0 | 1.3 |
| Slovakia | 30.0 | 25.8 | 4.2 |
| South Africa | 30.3 | 24.6 | 5.8 |
| South Korea | 32.0 | 25.1 | 6.9 |
| Spain | 31.7 | 24.5 | 7.3 |
| Sweden | 24.1 | 20.1 | 3.9 |
| Switzerland | 22.1 | 18.2 | 3.9 |
| Taiwan | 30.8 | 26.2 | 4.6 |
| Thailand | 24.2 | 20.9 | 3.4 |

(*Continued*)

**Table 3.** (Continued)

| Country | $\overline{SR^I}$ | $\overline{SR^{II}}$ | $\overline{SR^I} - \overline{SR^{II}}$ |
|---|---|---|---|
| Turkey | 33.2 | 25.2 | 7.9 |
| Ukraine | 39.5 | 36.9 | 2.6 |
| United Kingdom | 33.0 | 20.4 | 12.6 |
| United States | 56.1 | 21.7 | 34.4 |

Third, the difference between $SR^I$ and $SR^{II}$ varies over time differently depending on the country (Fig 4 and Table 2).

In particular, we can distinguish three groups of countries based on how the difference develops over time. The first group is characterized by an increasing difference, with the two curves of $SR^I$ and $SR^{II}$ progressively diverging. It includes Brazil, Chile, China, Colombia, Egypt, Hungary, India, Indonesia, Iran, Malaysia, Pakistan, Romania, Russian Federation, Slovakia, Turkey, and Ukraine. The second group shows a stable difference, meaning that the two curves follow parallel directions. It includes Argentina, Australia, Austria, Belgium, Canada, Croatia, Czech Republic, Denmark, Finland, France, Germany, Greece, Hong Kong, Ireland, Israel, Italy, Japan, Mexico, Netherlands, New Zealand, Norway, Poland, Portugal, Saudi Arabia, Singapore, South Africa, South Korea, Spain, Sweden, Switzerland, Taiwan, Thailand, and the United Kingdom. The last group includes only one country, the United States, where the

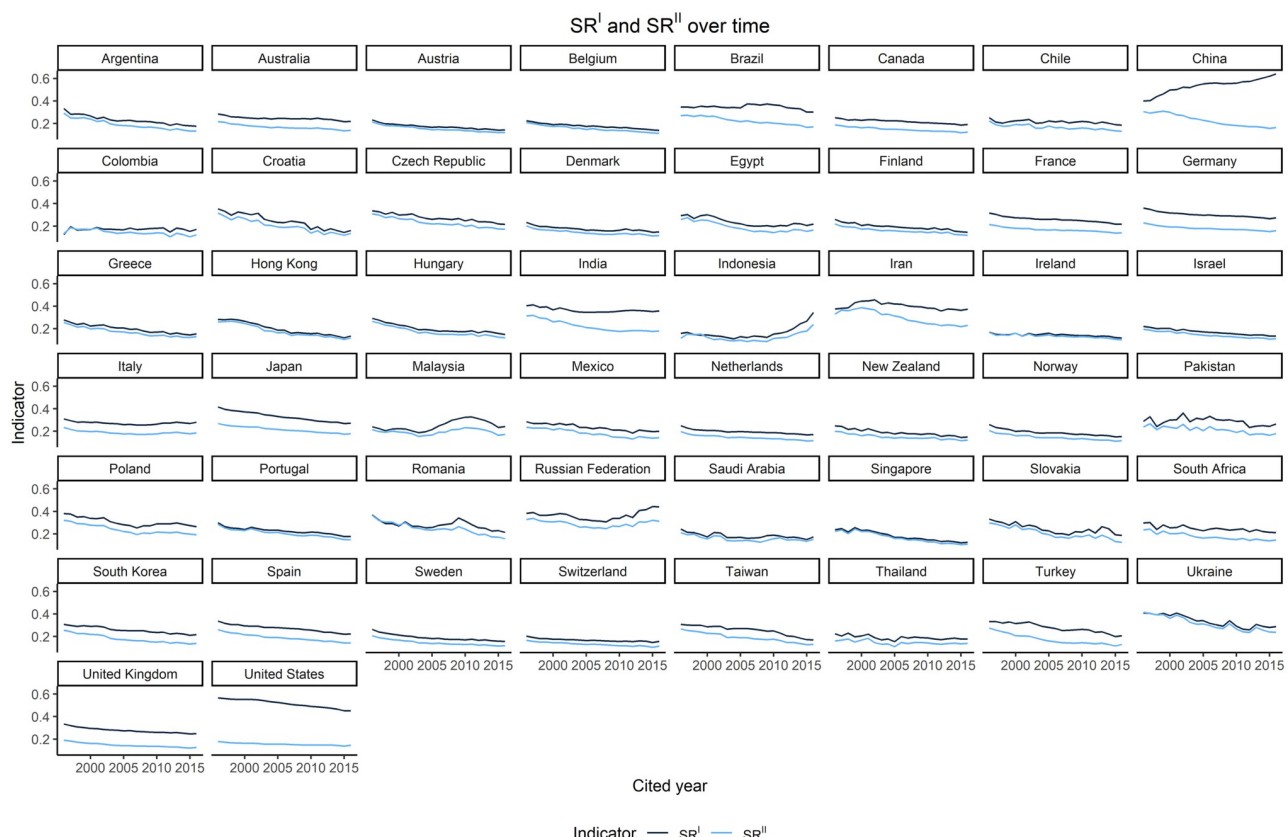

**Fig 3. Country self-citation rate $SR^I$ and $SR^{II}$ by country.** Yearly data 1996–2019.

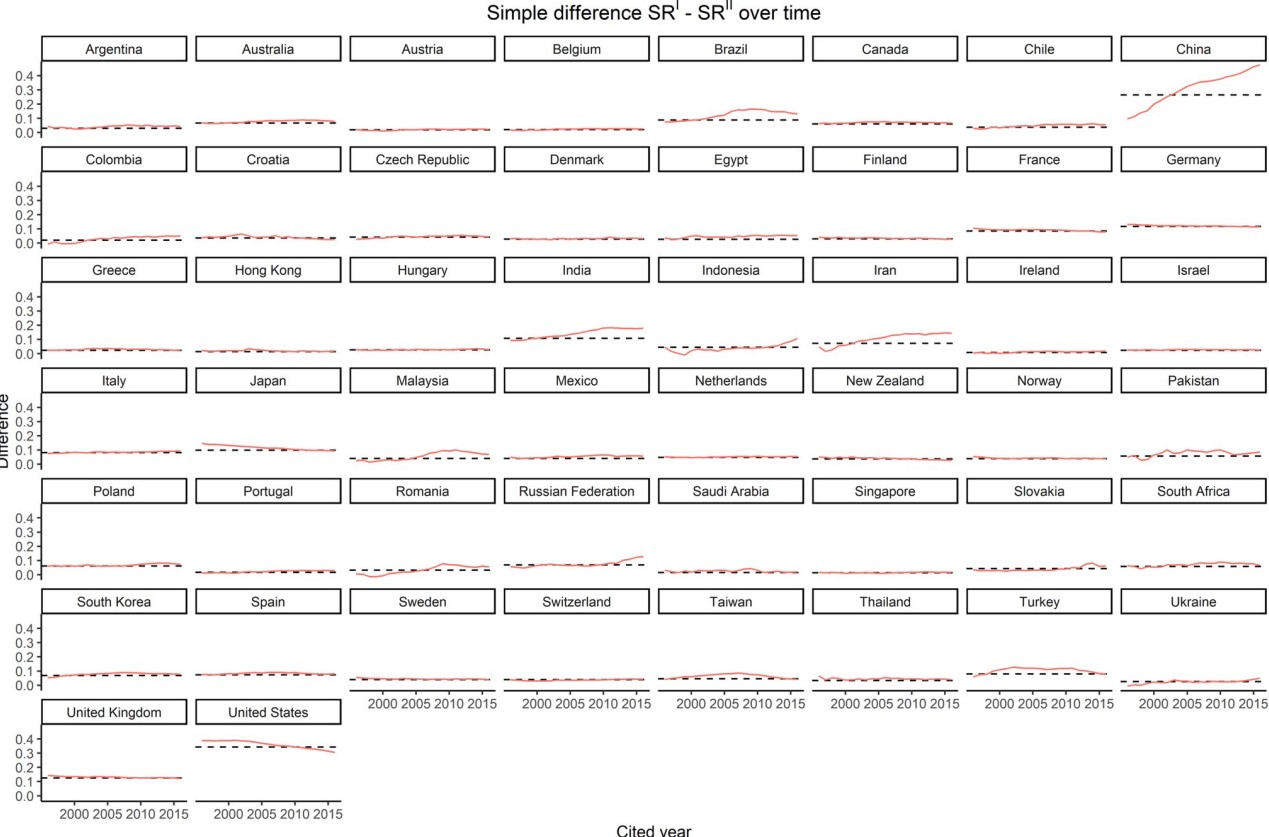

**Fig 4. Simple difference between $SR^I$ and $SR^{II}$ over time by country.** The solid red line represents yearly data (1996–2019), and the dotted black line is the average difference in the whole period.

difference reduces over time, i.e., the curves tend to converge. Note that all G10 countries, apart from the United States, belong to the second group and are all characterized by a significant difference between $SR^I$ and $SR^{II}$, with the United States showing the highest difference of all countries considered (mean difference = 34.4).

Both indicators result to be highly correlated with the logarithm of the publication output for most of the countries (Fig 5). Notably, the relationship between the self-citation rate (of both Types) and scientific production is of inverse proportionality: across most of the countries observed, an increase in scientific output is associated with a decrease in the self-citation rate. This contrasts with the direct proportionality found in prior studies [30, 40, 41, 46]. Interestingly, however, a few countries deviate from this general pattern, exhibiting either positive correlation (China for $SR^I$, Indonesia on both $SR^I$ and $SR^{II}$) or mild to low correlation coefficients (Colombia, Egypt, India, Iran, Italy, Malaysia, Pakistan, Romania, Russian Federation, Saudi Arabia, Thailand, Ukraine). The self-citation rate trend within this latter group of anomalous countries cannot be effectively predicted based solely on the trend of scientific production. (The scientific output trends of the 50 countries are reported in the Supporting Information).

To better investigate differences between groups of countries, the correlation-based distance was used to produce two matrices $M^I$ and $M^{II}$ of order $50 \times 50$, which represent the distances between the 50 countries' trends respectively on the $SR^I$ and $SR^{II}$.

The structures of the two matrices were visualized using Kruskal's Non-metric Multidimensional Scaling in the two MDS maps in Figs 6 and 7, based respectively on $SR^I$ and $SR^{II}$ trends.

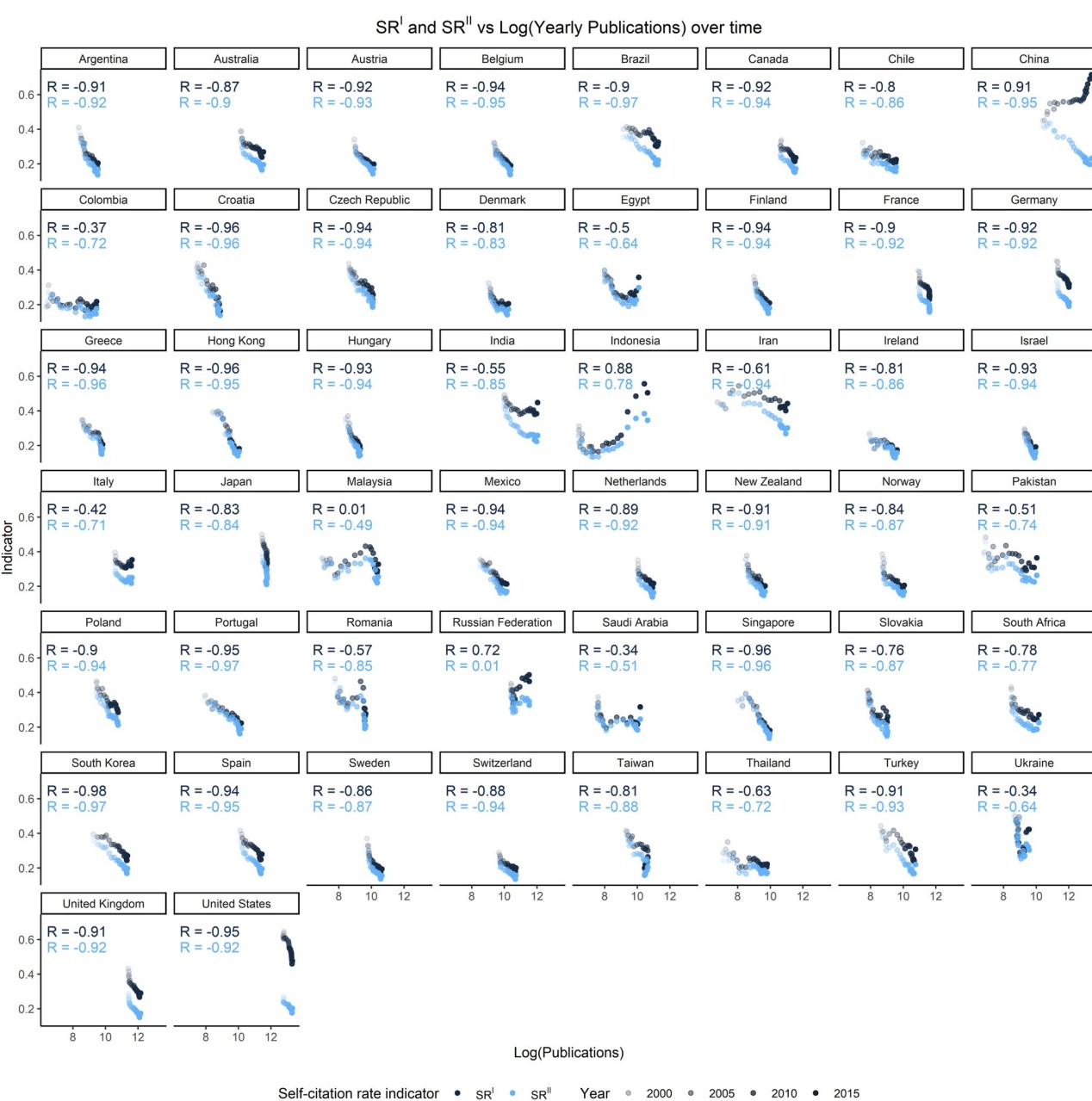

**Fig 5. $SR^I$ and $SR^{II}$ versus publication output by country.** Each data point represents data for a year between 1996 and 2019. Pearson correlation coefficients (by country and indicator) are overlaid.

As explained in Methods and data, the distance between the dots representing the countries on the MDS maps is inversely proportional to the similarity of their *SR* trends, so that countries characterized by similar trends will be placed closer and countries characterized by dissimilar trends far away. Note that, in both maps, the distances of the points on the 2-D map distort the original distances between the time-series only slightly, as shown by the low values of the stress of the MDS solutions (respectively, 9.18% and 7.45%). Note that the countries in the two latter zones are identical to those showing anomalous correlations with trends in scientific output.

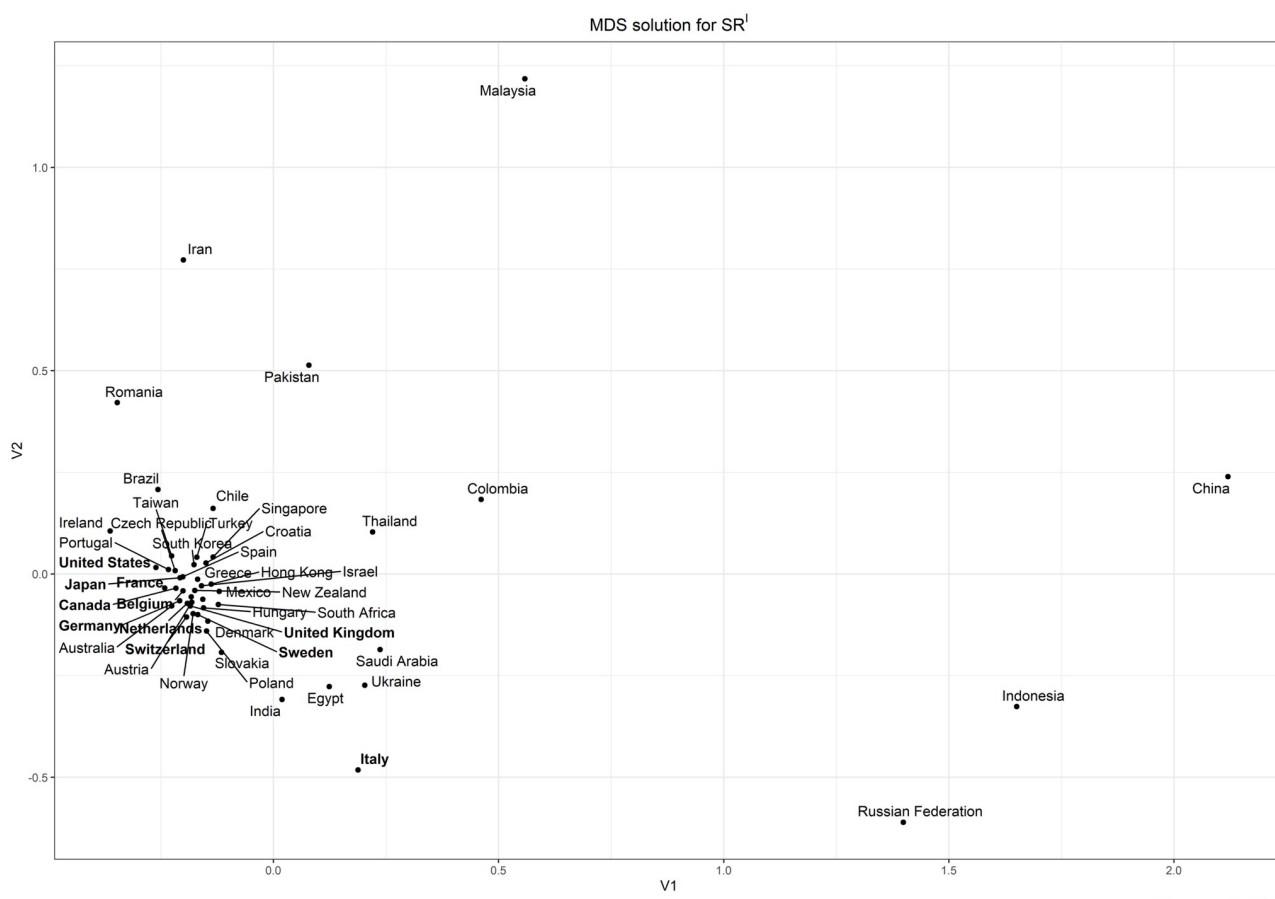

**Fig 6. MDS solution for $SR^I$.** G10 countries are in bold. Stress = 9.18%.

In the MDS map for $SR^I$ (Fig 6), three zones can be distinguished. The first is the big cluster in the left area, where most of the countries are concentrated. The second is the belt that surrounds the cluster and includes Iran, Romania, Pakistan, Thailand, Colombia, Saudi Arabia, Ukraine, Egypt, India, and, most notably, Italy, the only G10 country that is not placed inside the big cluster. The third zone comprises the rest of the map, where countries characterized by very specific trends are scattered: Malaysia, China, Indonesia, and the Russian Federation.

The MDS map for $SR^{II}$ (Fig 7) largely confirms this picture, showing significant overlap with the structural features of the previous map. Again, we find a big cluster including most of the countries, surrounded by a belt and, in the distant zones of the map, a scattering of anomalous countries. The belt includes the same countries as the belt in the previous map: Italy's anomalous position with respect to G10 countries is confirmed. The most important difference between the two maps is China: under the profile of the $SR^I$ trend, the country was placed in the distant zone, whereas under the profile of the $SR^{II}$ trend, it is placed within the big cluster.

## Discussion

The decreasing trend in both $SR^I$ and $SR^{II}$ which characterizes most of the countries shows that, in most of the cases, the overall citation impact of countries has grown more than the proportion of citation impact generated by domestic authors, i.e., that the denominator of both

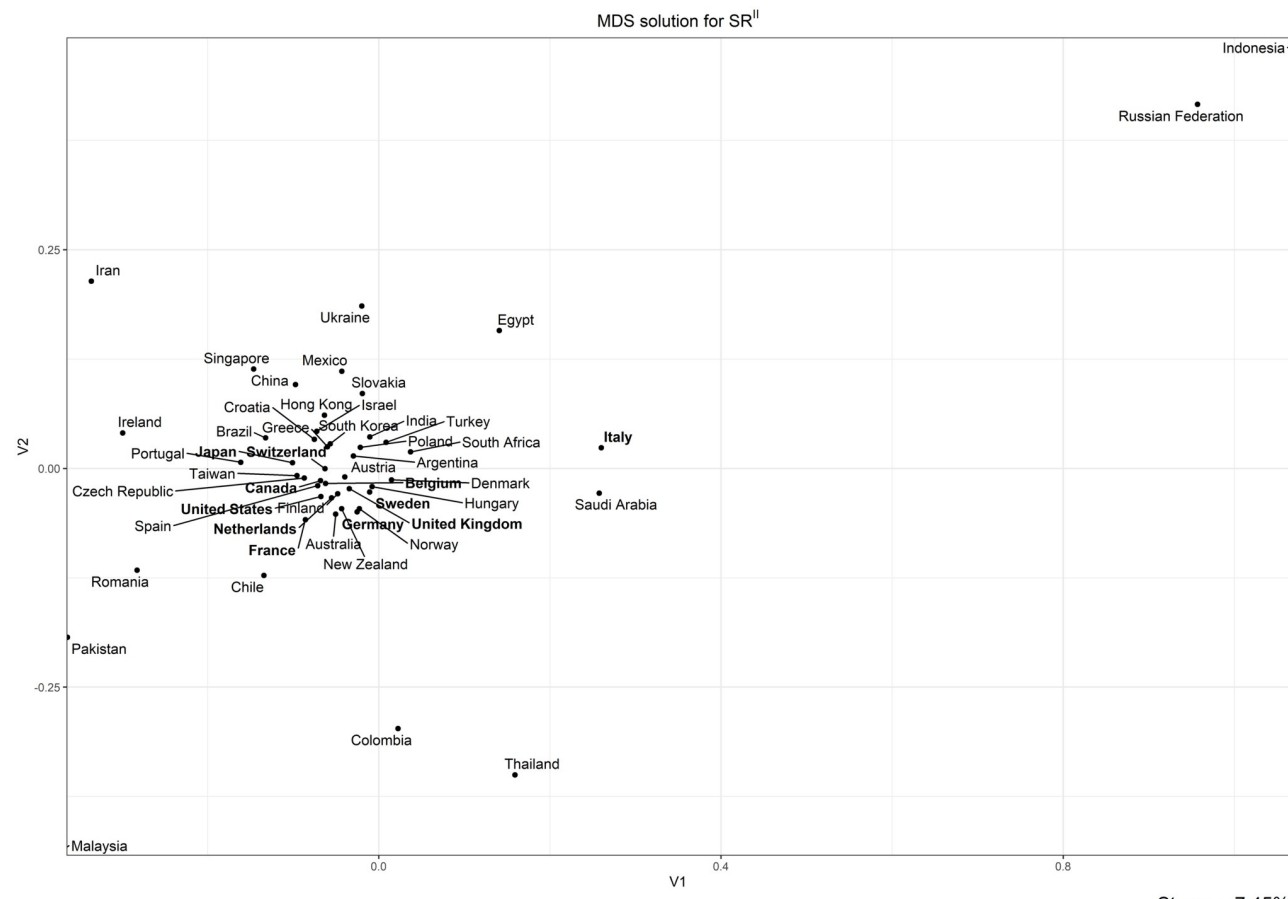

**Fig 7. MDS solution for $SR^{II}$.** G10 countries are in bold. Stress = 7.45%. Note that the points of Malaysia and Indonesia are outside the plot.

indicators has increased more than their numerator (see Section Indicator design). The faster increase in citations may be related in turn to the overall growth of scientific production and how it impacts the length of the reference lists of scientific publications. According to [57], in fact, the world's scientific production exhibits 4% annual growth in publications and 1.8% annual growth in the number of references per publication. Combined, these dynamics produce a 12-year doubling period in the total amount of references, which results in turn in a generalized increase in citations [58]. The decreasing trends, thus, may be simply due to the different rates of growth of the numerator and the denominator of the indicators used here: country self-citations of Type I or Type II grow less than bibliographic references.

The observed decreasing trend in the $SR^{I}$, however, contradicts previous studies of the development of this indicator over time: [26], in fact, report an average increase of +5.2 p.p. in the $SR^{I}$ of G10 countries between 2000 and 2016. Namely, the discrepancy between the present results and [26] depends on the different way of computing the $SR^{I}$ indicator. Indeed, [26] used a non-fixed citation window, which included all the years from the publication year to 2016. For example, for the cited items published in the year 2000, a 17-year citation window was used, by summing up all citations from 2000 to 2016; whereas, for the year 2006, the citation window was 11 years long, including citations from 2006 to 2016; for 2016, the citation window included only 1 year, i.e., only citations from 2016 itself were counted. Since self-

citations are in general younger than external citations [25, 59], they tend to represent a higher proportion of total citations for the years when the citation window is shorter. Hence, [26] registered inflation of $SR^I$ for more recent years, as the citation window shortens. The present study, by contrast, does not suffer from this problem as it is based on a fixed citation window, i.e., only a fixed number of years after the target year is considered (see Section Indicator design).

Turning now to a more substantial interpretation of the results, there are two patterns that are worth highlighting. First, the emerging giant of science, China, is characterized by a unique behavior: China $SR^I$ and $SR^{II}$ show almost opposite trends, with the former significantly increasing and the latter significantly decreasing. If the increasing $SR^I$ trend shows that Chinese scientists heavily rely on the scientific production of their own country, the decreasing $SR^{II}$ trend indicates that author self-citations are diminishing, in line with Western countries. Interestingly, this divergence results in a growing difference over time between the two indicators (Fig 4), which can be interpreted as a sign of the rising international impact of Chinese publications. As noted above, in fact, the difference between $SR^I$ and $SR^{II}$ depends on the citations of the international publications of a country. A wider difference means that these international collaborations are widely cited not only by the focal country (China, in our case) but also by the collaborating countries. Results, therefore, seem to show that international collaborations of Chinese authors are increasingly cited by other countries as well, another sign of the new status of China as a scientific superpower [60]. Notably, India shows an increasing difference as well, which may be interpreted analogously as a sign of the rising scientific impact of this country.

The second key pattern, emerging from the MDS maps, is that there are several countries whose self-citation behavior stands out from that of the majority of countries. With a couple of exceptions (China and India), these countries are the same when the two indicators are considered: Colombia, Egypt, Indonesia, Iran, Italy, Malaysia, Pakistan, Romania, Russian Federation, Saudi Arabia, Thailand, and Ukraine. Note that the anomalous self-citation rates of these countries cannot be explained by the trend of their scientific production, as evidenced by the low correlation coefficients found (Fig 5). This suggests searching for specific perturbations of the citation behavior.

Interestingly, all these anomalous countries have adopted, in the recent past, specific research policies aiming at increasing publication output and citation impact of their national scientific community. In the following sections, the recent history of the research policies in each group of anomalous countries is reconstructed. It will be shown that all these policies are characterized by direct or indirect incentives that may create room for the strategic use of self-citations. $SR^I$ and $SR^{II}$, therefore, seem to be sensitive to policy-induced perturbations of the citation habits.

## Post-socialist countries: Russian Federation, Ukraine, and Romania

Since 2007, Russian Federation adopted measures aimed at boosting research productivity, in the form of performance-based funding and individual payment for publications [61]. In 2012, Putin's May decrees N. 599 introduced various incentives for stimulating "the development of science in Russia and an increase in the number of articles by Russian scientists in the Web of Science Journals" [62, 63]. In particular, the project "5top100" aimed to push at least five Russian universities to enter the top hundred of leading international universities according to the global universities rank [64]. The project council paid attention to bibliometric indicators, including the number of publications and citations in Web of Science and Scopus [64]. At the local level, Russian universities introduced new promotion criteria and financial incentives for

faculty. After the policy intervention, the research productivity of the country significantly increased [63–67]. However, a contemporary rise in country self-citations in conference proceedings has been noted [63] and the spread of unethical practices after the policy change, including "predatory journals", plagiarism, and paper mills, has been repeatedly denounced [68, 69].

As to Ukraine, the new law "On scientific and Technical Activity" was enacted on January 16, 2016 (http://iht.univ.kiev.ua/ncst2016/index-en.html). It established a National Council of Ukraine on Science and Technology directly controlled by the ministers. The following year, the European Commission organized a Peer review of the Ukrainian Research and Innovation System largely supporting the new law. One of the key recommendations issued by the Commission was to identify research universities after a period of 5 years by taking into consideration also "the number of international publications and citations". In the meantime, the use of bibliometric indicators become largely diffused in Ukraine for ranking institutions [70], for distributing financial awards, and for evaluating projects [71]. According to [72], "until 2015 publication requirements for becoming associate professor and professor included only articles in Ukrainian journals. In 2015 they were substituted for articles in journals indexed in Scopus and WoS". According to scientists interviewed by [73], the Ukrainian academy suffers from a significant and inherited problem of misconduct and plagiarism. Possibly grafted on this tradition, new forms of adaptation to the bibliometric game are arising, such as publishing articles in selected national journals [74] and in Scopus de-listed journals [71]. Evidence about over-citations and self-citations has been provided as well [71].

Lastly, Romania started major reforms of tertiary education following the provisions of the Law of Education n. 1/2011 by modifying recruitment, university funding, and quality assurance [75]. It introduced also a research-driven classification and ranking system for universities managed by the Romanian Ministry of Education University, which constituted the informative basis of performance-based funding [76]. Academic and research staff recruitment and promotion changed radically from a model based on in-breeding to one taking into account individual performances measured by the number of publications and citations [75]. The Romanian Program for Rewarding Research Results, which had already started in 2007, was strengthened with direct payment to authors for publication in indexed journals [77]. [78] described in detail the functioning of the program, and highlighted that articles are rewarded according to the metrics of the journal where they are published. According to their analysis, monetary incentives supported productivity, but not the impact of Romanian research. The presence of incentives to publication might push toward misconduct, as suggested by the Romanian high level of retractions [79]. Evidence of self-citation and citation stacking for Romanian journals may be correlated to the necessity to boost journal metrics [80].

## Southeast Asia: Malaysia, Thailand, Indonesia

According to a UNESCO report [81], university rankings are central in the research policies of Malaysia and Thailand: "A key ingredient in high rankings is a university's publication rate. Consequently, faculty members—particularly those teaching in graduate programmes—are under pressure to publish in top-tier international journals". According to [82], Asian higher education institutions witnessed the "proliferation of policies surrounding the fanaticism with metrics [for] incentivizing scholars to publish through selected publication".

In Malaysia, the discussion about university performances and rankings started early [83]. Policy interventions happened in a highly centralized structure, where salaries and promotion criteria were defined directly by the Ministry of Education. In 2007, the Malaysian government adopted a National Higher Education Strategy Plan introducing performance-based funding

of universities [84]. The strategic objective to "empower research teams with new teamwork concepts to produce international level research output" was defined in an action plan for "improving the quality of faculty publications". Remarkably, the only indicator adopted was the "Increased percentage of staff achieving at least 100 citations" [85]. The upward trend of Malaysian country self-citations coincides with the years following this government intervention. In recent years, decentralized policies adopted by universities provide individual payment schemes for publications [86] (https://tinyurl.com/4t5kpd2h; https://tinyurl.com/mr33rzcn) and received citations (https://tinyurl.com/n3779dz3).

As to Thailand, the government initiated the establishment of the National Research University (NRU) project in 2009. "This plan aimed at developing academic excellences to enhance the country's research activities and to promote the better university-industry linkages for national competitiveness. The Office of the Higher Education Commission's selection criteria were mainly based on the ranking system conducted by Times Higher Education-Quacquarelli Symonds (THE-QS) and the impact factor of publications published on Scopus database" [87]. A list of excellent universities was also defined [88].

Indonesia was "the weakest nation in all relative scientometric indicators", with respect to Malayisia, Philippines, Thailand, and Vietnam [89]. According to [90] "the increase in the number of publications in recent years [. . .] is a reflection of government policy on research and academic careers and attempts to improve the position of Indonesian universities in international ranking". The main intervention was the Indonesian law on higher education Number 12/2012. [91] documented that "Academics who are successful in publishing their articles in Scopus- indexed journals would be rewarded with a certain amount of money and it goes directly to their pocket. Such a standard is also used as a measure for promotions [. . .], for payment of certified academics, and for honorary allowance. [. . .] The obligation to publish articles in reputable international journals has become integrated into doctoral programs and serves as a requirement to be met prior to the completion of the study". According to a survey [91], Indonesian "academics have attributed their interpretation of the rewards as a mere completion of publishing in any kinds of journals indexed in Scopus apart from the consideration of the quality. Consequently, academics have performed their own way or strategy of publishing through the easiest, fastest, cheapest open access journals and proceedings". The performance in terms of citations per paper appeared not so high with respect to other neighborhood countries until 2017 [92]. [93] documented that Indonesia has made a recent and remarkable shift towards conference proceedings publishing. Rochmyaningsih [94] criticized the adoption of this aggressive policy by arguing that "the developing world needs more than numbers". In 2017 Indonesia's Ministry of Research, Technology and Higher Education introduced Indonesia's Science and Technology Index (SINTA; https://sinta.kemdikbud.go.id/ ), based on Scopus data, that "gives recognition to Indonesian scientists, triggers competition among them, and motivates them to be better" [95]. The number of papers, citations, and H-index are mixed in an index used for evaluating research grants applications, promotions, and salary negotiations. According to [95] several top scorers "had inflated their SINTA score by publishing large numbers of papers in low-quality journals, citing their own work excessively, or forming networks of scientists who cited each other". The present paper documents that the adoption of policies both in 2012 and 2017 was flanked by a modification of self-citation strategies of Indonesian scientists.

## Muslim-majority countries: Egypt, Iran, Saudi Arabia, Pakistan

According to [96], Egypt, Iran, Pakistan, and Saudi Arabia differ from other Muslim-majority countries in terms of research performance. In particular, Egypt and Saudi Arabia have been

the most active research producers from the Arab world in the last 20 years. Like other Arab countries, they adopted a reform of tertiary education and witnessed a remarkable growth of publications and citations [96–98], matched however by a rising number of retractions due to misconduct [99, 100]. More in general, according to [101], the problem of corruption is widespread in Arab universities.

Since the mid-2000s, Saudi Arabia adopted National Development plans aiming to support research productivity [102], by mixing centralized strategies, such as the National Science Technology and Innovation Plan inspired by the US National Science Foundation, and decentralized ones adopted by universities [103]. According to [104] this catching-up strategy of Saudi Arabia universities started in 2007, and it was mainly based on attempts to raise research outputs, prestige, and rankings (e.g. https://tinyurl.com/4s8b6hef), by allocating "significant research funding to support industry-based Research Chairs as well as the employment of high-profile international researchers to lead projects that will be staffed by university faculty and postdoctoral students" [102]. Individual incentives for researchers are widely adopted, with salary increases and promotions tied to publications and citations (e.g. https://tinyurl.com/2s3zchwk; https://tinyurl.com/5n96nmrt). The strategy of affiliating to Saudi Arabia universities external highly cited researchers for improving rankings received early criticism [105–107]. Nonetheless, [108] claims that self-citations are not anomalous in Saudi Arabia, at least in the medical specialties.

In comparison to Saudi Arabia, the governance of Egyptian universities is traditionally much more centralized [109]. According to [98], Egypt "demonstrate[s] the importance of incentives within hiring organizations, specifically whether researchers are rewarded for publications or obtaining funding". Indeed, the Ministry of Scientific Research introduced competitive funding to research in 2007, by favoring basic research, the formation of research groups, and international collaborations. At a single university level, "internal practices recognize and reward certain forms of performance more than others—such as teaching, administration, graduate supervision, advising and outreach—as well as the expected quantity and prestige of scientific publications" [98]. According to some researchers interviewed in [110], individual financial incentives and national awards [111] represent the main push leading to the improvement of Egypt's higher education sector. Others have a less positive attitude and highlight the diffusion of malpractices in research such as plagiarism, data fabrication, and manipulation [112]. According to a comprehensive survey of Egyptian researchers, financial rewards for publications together with low salaries are among the most important risk factors leading to research misconduct [112]. Finally, it should be noted that Egyptian universities provide individual awards for citations (e.g. https://bu.edu.eg/BUNews/25947).

Regarding Pakistan, in 2002, the country's government established a Higher Education Commission in alignment with the World Bank's task force recommendations. This commission was aimed at expanding the higher education sector and improving research within the country. Various measures were adopted, including incentives aimed to promote research and scholarship [113–115]. Since 2002, a financial incentive based on the number of publications, number of citations received, and Impact Factor of journals was also introduced [116, 117]. The growth of international publications, collaborations, and citations is considered a result of these policies [114, 117, 118]. Still, Pakistan is currently one of the leading countries in terms of retractions [79, 99], and, according to [119], under the rising menace of scholarly black-market pushed-up by monetary incentives.

Iran had between 1980 and 2010, "one of the fastest rates of growth in scientific production that the world has witnessed", probably due to the nuclear technology development program [120]. In 2009, Iran announced a 20-year "comprehensive plan for science" focused on higher education and stronger links between industry and academia [121, 122]. The quantitative

growth of Iranian science has continued until now [121, 123]. It is a controversial issue, however, whether this development has been matched by the increase of scientific quality too [123, 124]. According to a critic, "The state has imposed deeply short-sighted research policies on universities with the sole objective of increasing the number of publications, which is in turn used in its propaganda to demonstrate progress in technological self-sufficiency and mask significant shortcomings caused by decades of isolation due to the regime's international policy. [. . .] A top-down incentive for publication along with lack of real demand from the economy, which is not based on new technology development, have pushed Iranian researchers to focus on the publishability of their works rather than their relevance and practical impact" [124]. As in the other countries, "the government's policy in higher education makes academic promotions and student graduation contingent upon publication of papers in scientific journals. These policies have created an environment that lends itself to the most extreme form of the publish-or-perish paradigm" [125]. Retractions of articles authored by Iranian scientists have attracted attention worldwide [126, 127]. An anomalous rate of self-citations has been documented as well [128].

## Colombia

Since 2002, Colombia has introduced a model of wage incentive based on research productivity, the Faculty Promotion Policy for Colombian Public Universities (Decree 1279 of 2002). The performance rating is based on a national index of scientific journals called Publindex. In 2009, the Ministry of Education started to measure production in terms of citations in the WoS database. In 2018, the system was finally strengthened [129]. By and large, this legislation established a pay-for-performance system through salary points calculated in accordance with the higher education degrees, academic rank (fixed components), and academic productivity (variable component) [130–132]. [129] documented by anecdotal evidence that self-citations and citation clubs are perceived as current problems by Colombian scholars.

## Italy

Italy is among the top 10 producers of science in terms of global production and total citations (Table 2) and it has been considered the main European competitor of the United Kingdom for citation impact [133, 134]. However, data shows that Italy is the only G10 country exhibiting an anomalous trend of self-citations. The Italian anomaly is especially visible when compared with the trends of the other G10 countries: from 2010, Italy starts to diverge from the benchmark countries in both indicators (Fig 8). At the end of the observation period, it results to be the country with the highest $SR^{II}$, above Japan.

The beginning of the change in the trend coincides with the wide process of reformation of the Italian university system started by the government in 2010 (Law 240/2010). A governmental agency (ANVUR) was established in charge of monitoring and evaluating the Italian research system and, in 2011, the first national research assessment exercise started, followed by a second round in 2015. In both, the evaluation was largely based on the automatic or semi-automatic use of algorithms fed by citation indicators [135, 136]. Universities started to be funded according to their performance in the research assessments. Moreover, the reform modified also the recruitment and advancement system for university professors by introducing a national scientific qualification (ASN). This qualification is mandatory both for hiring and promotion and, in order to obtain it, candidates in natural sciences, life sciences, and engineering must exceed publication and citation thresholds centrally defined by ANVUR [26]. As a result of these reforms, bibliometric performance has acquired a central role in the career of

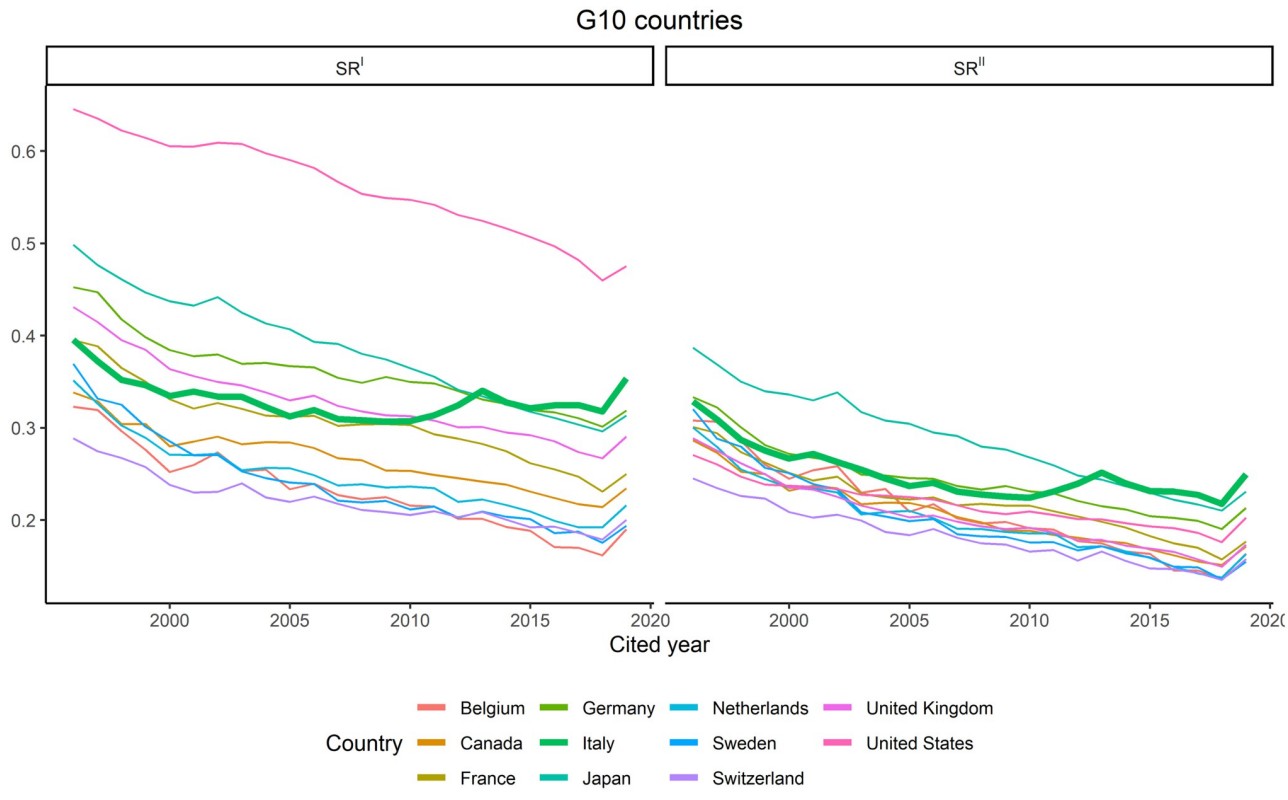

**Fig 8. G10 countries' trends in $SR^I$ and $SR^{II}$.** Italian trends are in bold green.

Italian scientists [137]. At the same time, anomalous rises in Italian self-citations have been documented by several studies [26, 34–38].

## Conclusions

Since the early times of citation indexes, preoccupations with the opportunistic use of citations have been voiced by bibliometricians and the scientific community [138, 139]. The centrality acquired by metrics in the various ganglia of the research system, from the career of individual scientists to the ranking of institutions until the evaluation of the scientific performance of entire countries, has further sharpened these concerns [6, 140–143]. Self-citation, in particular, has persistently been indicated as among the easiest strategies available to scientists for artificially boosting their citation-related performance indicators [6, 144], raising the question of whether scientists, under pressure, do indeed attempt to game citation scores by increasing self-citation.

In the present study, we investigated how the propensity to self-citation changed in 50 countries all over the world from 1996 to 2019, using two different indicators based on country self-citations. The results show that, for most countries, self-citation rates have decreased over time following similar patterns. Tendency to self-citation, both at the country level and for individual scientists, seems in fact to be declining. Against this background, however, there are some countries that exhibit anomalous behavior, showing self-citation trends that are significantly different from those of "standard" countries.

The analysis of the research policies adopted by these anomalous countries in the last years has revealed that they *all* share a common trait, namely the introduction of direct or indirect

rewards for the bibliometric performance of scientists. The temporal association we found in all anomalous countries between changes in policies on the one hand and changes in the self-citation behavior of the national scientific community on the other hand suggests that scientists do indeed respond to the new climate of incentives by altering, among other things, their citation habits. Policy pressure seems therefore capable of affecting rapidly and visibly the citation behavior of entire countries, possibly distorting global rankings of countries based on citations [31].

Clearly, we cannot offer a full-fledged *causal* explanation of our data, displaying the causal chains that start from the policy and end with the citation choices of individual authors. Neither we can demonstrate that the *whole* self-citation gain of anomalous countries is due to opportunistic adaptation to research policies. Still, the most likely high-level explanation of the macro-trends we observe is that the policies centered on or emphasizing citation performance do significantly affect the behavior of scientists.

Remarkably, our results show that merely having a performance-based funding system involving citation-based indicators is insufficient for a country to exhibit anomalous self-citation behavior. For example, countries like Belgium, Bulgaria, Croatia, and Sweden employ citation-based indicators in evaluating their university systems, yet they do not display anomalous self-citation tendencies [145]. It appears that the crucial factor influencing this anomalous self-citation behavior is the *proximity* of incentives based on citations to an individual researcher's career and wage. It seems that the closer these incentives are to affecting researchers' career progression and wages, the more likely they are to influence citation behavior. In Italy, for instance, the national scientific qualification, which is awarded based on citation indicators, has a direct impact on the career of researchers, as without it researchers cannot access tenure-track positions or advance to full professorships. By contrast, in countries where citation-based indicators contribute to complex funding formulas at the university level, the influence on citation behavior is more distant from individual researchers and, consequently, less significant.

In future research, it will be important to further investigate how varying architectures of performance-based funding systems have more or less impact on citation behavior, exploring in particular the extent to which the proximity of citation-based incentives to individual careers and wages induces strategic behavior. Other promising topics for future research include examining whether research fields characterized by varying epistemological, social, and institutional features respond differently to science policies and investigating whether citation incentives may affect countries' mix of scientific output, inducing specialization in areas rewarded by citation metrics (e.g., the medical sciences) at the expense of less "performing" areas (e.g., the humanities).

On its part, this study contributes to the ongoing discussion on research evaluation systems by showing that when bibliometric indicators, and citation-based indicators in particular, are integrated into systems of incentives, they cease to be neutral measures to become active components in the research system. As such, they are able to modify the behavior of entire scientific communities. Hence, they should be handled by science policy makers with the most extreme caution.

## Supporting information

**S1 File. $SR^I$ and $SR^{II}$ trends using 1-year and 5-year citation windows; MDS solutions for $SR^I$ and $SR^{II}$ using 1-year and 5-year citation windows; publication output of the 50 countries between 1996 and 2019.**
(PDF)

**S1 Data. Raw data used for computing the indicator.** Further indicators computed: over-citation ratio, odds-ratio, and relative self-citation rate.
(CSV)

## Acknowledgments

This work uses Scopus data provided by Elsevier through the ICSR Lab. We are very grateful to Lucio Barabesi, Jeroen Bass for his help in retrieving the data, and Giuseppe De Nicolao for insightful discussions in the early stage of the research.

## Author Contributions

**Conceptualization:** Alberto Baccini, Eugenio Petrovich.

**Data curation:** Alberto Baccini, Eugenio Petrovich.

**Formal analysis:** Alberto Baccini, Eugenio Petrovich.

**Funding acquisition:** Alberto Baccini.

**Investigation:** Alberto Baccini, Eugenio Petrovich.

**Methodology:** Alberto Baccini, Eugenio Petrovich.

**Supervision:** Alberto Baccini.

**Validation:** Alberto Baccini, Eugenio Petrovich.

**Visualization:** Alberto Baccini, Eugenio Petrovich.

**Writing – original draft:** Alberto Baccini, Eugenio Petrovich.

**Writing – review & editing:** Alberto Baccini, Eugenio Petrovich.

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
