## [Decision Letter · Decision Letter 0]

15 Aug 2023

PONE-D-23-07697A global exploratory comparison of country self-citations 1996-2019PLOS ONE

Dear Dr. Baccini,

Thank you for submitting your manuscript to PLOS ONE. After careful consideration, we feel that it has merit but does not fully meet PLOS ONE’s publication criteria as it currently stands. Therefore, we invite you to submit a revised version of the manuscript that addresses the points raised during the review process.

We look forward to receiving your revised manuscript.

Kind regards,

Mu-Hsuan Huang

Academic Editor

PLOS ONE

Journal Requirements:

“This work uses Scopus data provided by Elsevier through the ICSR Lab. We are very 779 grateful to Lucio Barabesi, Jeroen Bass for his help in retrieving the data, and Giuseppe 780 De Nicolao for insightful discussions in the early stage of the research. Alberto Baccini 781 acknowledges the funding by the Italian Ministry of University, PRIN project: 782 2017MPXW98”

“Alberto Baccini acknowledges the funding by the Italian Ministry of University, PRIN project: 2017MPXW98”

Reviewers' comments:

Reviewer's Responses to Questions

**Comments to the Author**

1. Is the manuscript technically sound, and do the data support the conclusions?

Reviewer #1: Yes

Reviewer #2: No

2. Has the statistical analysis been performed appropriately and rigorously? 

Reviewer #1: Yes

Reviewer #2: No

3. Have the authors made all data underlying the findings in their manuscript fully available?

Reviewer #1: Yes

Reviewer #2: No

4. Is the manuscript presented in an intelligible fashion and written in standard English?

Reviewer #1: Yes

Reviewer #2: Yes

5. Review Comments to the Author

Reviewer #1: The issue is relevant. Indeed, the centrality of metrics in the research system, from individual researchers' careers to institutional rankings and country assessments, has heightened these concerns. The study examines changes in self-citation rates in 50 countries from 1996 to 2019 and finds that, overall, self-citation trends appear to be declining. However, some countries show anomalous behaviour, with self-citation trends that differ significantly from the "standard" countries. The article is well presented and respects the structure of a scientific article. The analysis focuses on a phenomenon of self-citation with a discussion of bibliometrics at the meso level. The discussion focuses on science policy, in line with the problem.

It is argued that policies focused on citation performance significantly affect the behaviour of scientists and calls for caution in the use of bibliometric indicators, as they can become active components of the research system and change the behaviour of scientific communities at the country level.

The state of the art allows the definition of self-citation and self-reference. It also presents the formal framework within which the study is carried out.

The method is based on the analysis of time series with a differentiation between countries in the context of self-citation, which allows a more detailed analysis of this issue. The literature is well used in the construction of the method. Figure 1, although brief, contributes to a good understanding of the article. The section on design indicators is explicit about implementation.

A sensitive part of the study concerns the comparison of time series. The chosen approach allows the calculation of an adaptive dissimilarity index between two time series. This makes it possible to capture both the raw dissimilarity and the dissimilarity in the temporal correlation behaviour.

The figures could be improved. Figures 3 and 4 and the legend need to be enlarged to make them easier to read. The text of Figure 5 should be enlarged.

In the discussion, the presentation of the different policies per country is relevant.

The conclusion is well balanced with the limitations of the study.

The results obtained and the discussion invite a more detailed conclusion on the limitations and perspectives of this study, in order to offer a deeper reflection on self-citation in a global context and on its nature.

The bibliography is well constructed and used. DOIs are present.

Some typographical and spelling errors need to be corrected and require a second reading: l 129, l 436, l 574, l 132, ...

Reviewer #2: The main purpose of this work is described in line 43 of the Introduction: “In this study, the trends of country self-citations in 50 countries over the world are investigated in order to reveal groups of countries characterized by similar self-citation behavior in time”.

Self-citations are a key topic in evaluative bibliometrics. Country self-citations are a very useful indicator at the macro level to highlight certain phenomena that can be traced to structural elements of a country's research infrastructure or to national policies adopted within it.

The topic addressed is, therefore, extremely relevant and highly topical. However, although well written, easy and clear to understand, the paper has some remarkable methodological weaknesses which, unfortunately, undermine its results and therefore fail to substantiate the findings.

My major concern about the methodological framework adopted by the authors is represented by the choice to use self-citation and not self-referencing.

Already in the abstract, the authors state that: “12 countries (…) exhibit anomalous trends of self-citations. We argue that these anomalies should be attributed to the aggressive science policies adopted by these countries in recent years, which are all characterized by direct or indirect incentives for citations. Our analysis confirms that when bibliometric indicators are integrated into systems of incentives, they are capable of affecting rapidly and visibly the citation behavior of entire countries”.

In line 513 of the Discussion: “All these policies are characterized by direct or indirect incentives that may create room for the strategic use of self-citations. SRI and SRII (the two indicators on which the whole analysis is based), therefore, seem to be sensitive to policy-induced perturbations of the citation habits”.

The point is that “Habits” refers to “active” and not “passive” behaviours, therefore, their analysis cannot be based on self-citations but on self-references. Note that the word “behaviour” appears 19 times in the paper (5 in the Conclusions).

I think it is a very serious fault also because the authors do not deny this evidence, and at the very beginning (line 7 of the introduction) they say, “As for author self-citations, also country self-citations should be distinguished from country self-references”. “Notably, country self-references depend only on the behavior of the researchers from the country. By contrast, country self-citations are determined also by the behavior of other citing countries”.

How can the authors investigate the possible occurrence of "opportunistic behavior" through the use of an indicator tracking the behavior of entities not subject to the "treatment" (the incentive)?

More importantly, how can such a choice be justified considering that instead, there is an indicator that tracks the behavior of all and only those entities subject to the "treatment"?

Even more difficult to understand the choice to adopt self-citations instead of self-references, considering the computational complications involved. Line 69: “The computation of self-citations requires the definition of: (i) a publication window …; b) a citation window”. Line 103: “The computation of self-reference requires only a publication window for the set of citing sources”. And also, in line 321: “A citation window is required to correct for the fact that older publications have more time to accumulate citations”. Not required if self-references are used in place of self-citations.

In summary, I believe that every sentence in the paper should be reconsidered in light of this "original sin."

For instance, “The country self-citation rate results to be positively correlated with the publication output of a country”. I think the correct statement is, "The country self-reference citation rate results to be ...". The more output a country produces, the more chance the country has to self-cite.

Or: “The odds-ratio ON,y measures to what extent the country relative preference to cite its own publications”. Wrong: this refers to self-referencing.

Even more in the Conclusions, two examples for all:

• “Self-citation, in particular, has persistently been indicated as among the easiest strategies available to scientists for artificially boosting their citation-related performance indicators, raising the question of whether scientists, under pressure, do indeed attempt to game citation scores by increasing self-citation”.

• “Scientists do indeed respond to the new climate of incentives by altering, among other things, their citation habits. Policy pressure seems therefore capable of affecting rapidly and visibly the citation behavior of entire countries”.

Other points:

1) “A citation is considered a country self-citation when the intersection between the set of the countries of affiliation of the author(s) of the cited publication and the set of the countries of affiliation of the author(s) of the citing publication, is not empty”.

This may be acceptable only if the self-citation is attributed solely and exclusively to the “intersection” and not to all authors' countries. As a consequence, one of the two indicators used by the authors (country self-citations of Type I) is wrong, being affected by systematic error.

Among others, the authors use this indicator in order to capture “the effect of citation clubs at the country level” … really suggestive, but honestly, at least a bit far-fetched to be achieved with such an indicator.

2) “In static analyses of country self-citations, indicators encapsulating both publications and country self-citations are useful, but they become problematic when the temporal dimension is considered”. [Therefore] … the most suitable indicator is the self-citation ratio only”.

This is true, but if you want to capture truly anomalous events (possible strategic behaviours), you must necessarily analyse self-citations together with output. As stated by the authors, if there is a clear causal nexus between output and self-citations, then an "entity" E with an anomalous pattern in terms of output will be able to register, ceteris paribus, an anomalous pattern in terms of self-citations as well. This applies to individual scholars and also to countries.

3) Again, about the causal link between output and self-citations, it is clear that the authors' interpretation of anomalous patterns on self-citations, I guess, applies to output as well. Are we sure that this is “citation gaming”? Doesn't it all depend on the expansion of the output of these countries? The methodological approach proposed by the authors is inadequate to answer that.

4) “When bibliometric indicators are integrated into systems of incentives, they cease to be neutral measures to become active components in the research system. As such, they are able to modify the behavior of entire scientific communities. Hence, they should be handled by science policy makers with the most extreme caution”. Correct and agreeable statement. But then again, it MUST be so, for any metric implemented in an organised system to incentivise some behaviours (e.g., productivity increase), even with the awareness that this may induce collateral effects.

5) Table 2 caption: "Citing sources include all types of documents." What about cited ones? Wouldn't it be worth limiting the analysis to journal articles only?

6) What about field dependency of citation distributions? It is not real to assume that all the top 50 analysed countries are equal in terms of field specialisations. As for the intertemporal analysis, if a country's mix of scientific (input) output changes (perhaps because of that country's policy), the overall citation pattern also changes. At least one brief comment on this limitation should be inserted in the paper.

6. PLOS authors have the option to publish the peer review history of their article (what does this mean?). If published, this will include your full peer review and any attached files.

Reviewer #1: No

Reviewer #2: No

---

## [Author Response · Author response to Decision Letter 0]

5 Sep 2023

As requested by the editorial office, we dropped funding information from the acknowledgements and we inserted it in the funding statement.

---

## [Decision Letter · Decision Letter 1]

2 Oct 2023

PONE-D-23-07697R1A global exploratory comparison of country self-citations 1996-2019PLOS ONE

Dear Dr. Baccini,

Thank you for submitting your manuscript to PLOS ONE. After careful consideration, we feel that it has merit but does not fully meet PLOS ONE’s publication criteria as it currently stands. Therefore, we invite you to submit a revised version of the manuscript that addresses the points raised during the review process.

We look forward to receiving your revised manuscript.

Kind regards,

Mu-Hsuan Huang

Academic Editor

PLOS ONE

Journal Requirements:

Reviewers' comments:

Reviewer's Responses to Questions

**Comments to the Author**

1. If the authors have adequately addressed your comments raised in a previous round of review and you feel that this manuscript is now acceptable for publication, you may indicate that here to bypass the “Comments to the Author” section, enter your conflict of interest statement in the “Confidential to Editor” section, and submit your "Accept" recommendation.

Reviewer #1: All comments have been addressed

Reviewer #2: (No Response)

Reviewer #3: (No Response)

2. Is the manuscript technically sound, and do the data support the conclusions?

Reviewer #1: Yes

Reviewer #2: No

Reviewer #3: Yes

3. Has the statistical analysis been performed appropriately and rigorously? 

Reviewer #1: I Don't Know

Reviewer #2: No

Reviewer #3: Yes

4. Have the authors made all data underlying the findings in their manuscript fully available?

Reviewer #1: Yes

Reviewer #2: Yes

Reviewer #3: Yes

5. Is the manuscript presented in an intelligible fashion and written in standard English?

Reviewer #1: Yes

Reviewer #2: Yes

Reviewer #3: Yes

6. Review Comments to the Author

Reviewer #1: The response produced by the authors has taken account of the many criticisms levelled by the reviewers, both in terms of form and content.

The authors have produced new results to answer the questions raised by the reviewers.

raised by the reviewers.

They have also produced a more detailed explanation of their approach,

and less polemic around the issue of self-citation and self-referencing.

Nevertheless, there are still divergent points of view on this study.

The authors' response seems to me to be part of the scientific debate,

It is for this reason that I agree to the publication of this paper.

It is important that the data from this study be made available

for reproducibility and study purposes, in order to continue the scientific debate

which is in the process of being established.

Reviewer #2: The authors' replies are unsatisfactory and, for this reason, I'm suggesting rejection of the manuscript.

My comments are in the attached file.

Reviewer #3: The present manuscript is an investigation into temporal developments into country self-citations, especially in relation to science policies at national level which involve citation metrics. Self-citations are simple but also easily observed means to strategically influence citation metrics. This study provides and extensive review of country-level self-citation methodology and continues to report on country-level values. The study is overall well designed and conducted. I have only a few remarks to the authors to possibly improve their paper.

In their choice of two country self-citation indicators, the rationale for the Type II indicator is completely clear but that for the Type I indicator is not. This indicator's definition and logic seem to contradict the very meaning of the term "country self-citations" even though it involves self-citations and countries. In this indicator, as I understand it, all co-authors' citations of a target paper are considered. For all countries involved in the target paper, any co-author self-citation is counted, regardless of that author's country. This means both same-country author self-citations and different-country self-citation are counted, contrary to the Type II indicator. I do not follow how a different country's author self-citations can possibly be called country self-citations.

Regarding the discussion of results: if the countries identified as anomalous in terms of self-citation indicators "all share a common trait, namely the introduction of direct or indirect rewards for the bibliometric performance of scientists" (p. 27), is the opposite also true? Namely, is it the case that all the other countries do not have national policies involving bibliometrics? This is an important further consideration. If there are such countries with citation-based incentives and they don't show anomalous country self-citation developments, this would seriously change the interpretation of results.

Finally, the reference list of this paper is poorly quality checked and should be very carefully revised. Only looking superficially I found the following faulty references and there may well be more:

69. Abalkina A. type [; 2021]Available from:

https://scholarlykitchen.sspnet.org/2021/02/04/

guest-post-unethical-practices-in-research-and-publishing-evidence-from-

83. Gill J. type [; 2008]Available from: https://www.timeshighereducation.com/

news/malaysian-rankings-flop-shames-the-nation/404570.article.

85. Malaysia MoEo. type [; 2007]Available from: https://www.ilo.org/dyn/

youthpol/en/equest.fileutils.dochandle?p_uploaded_file_id=477.

113. ur Rahman A. Building a Knowledge Economy. In: Baydoun E, Hillman JR,

editors. Universities in Arab Countries: An Urgent Need for Change:

Underpinning the Transition to a Peaceful and Prosperous Future. Cham:

Springer International Publishing; 2018. p. 105–121. Available from:

https://doi.org/10.1007/978-3-319-73111-7_5.

134. for Science Innovation D, Technology U. type [; 2022].

These issues notwithstanding, I think the paper already meets the criteria for being published. I would thus recommend that the authors seriously consider them.

7. PLOS authors have the option to publish the peer review history of their article (what does this mean?). If published, this will include your full peer review and any attached files.

Reviewer #1: No

Reviewer #2: No

Reviewer #3: No

---

## [Editor Report · Decision Letter 2]

7 Nov 2023

A global exploratory comparison of country self-citations 1996-2019

PONE-D-23-07697R2

Dear Dr. Baccini,

We’re pleased to inform you that your manuscript has been judged scientifically suitable for publication and will be formally accepted for publication once it meets all outstanding technical requirements.

Kind regards,

Mu-Hsuan Huang

Academic Editor

PLOS ONE

---

## [Editor Report · Acceptance letter]

10 Nov 2023

PONE-D-23-07697R2 

A global exploratory comparison of country self-citations 1996-2019 

Dear Dr. Baccini:

I'm pleased to inform you that your manuscript has been deemed suitable for publication in PLOS ONE. Congratulations! Your manuscript is now with our production department. 

Kind regards, 

on behalf of

Professor Mu-Hsuan Huang 

Academic Editor

PLOS ONE